# Simultaneous brain, brainstem, and spinal cord pharmacological-fMRI reveals involvement of an endogenous opioid network in attentional analgesia

Valeria Oliva[1,2,3], Ron Hartley-Davies[2,4], Rosalyn Moran[5], Anthony E Pickering[1]*, Jonathan CW Brooks[2,6]

[1]Anaesthesia, Pain & Critical Care Sciences, School of Physiology, Pharmacology and Neuroscience, Biomedical Sciences Building, University of Bristol, Bristol, United Kingdom; [2]Clinical Research and Imaging Centre, School of Psychological Science, University of Bristol, Bristol, United Kingdom; [3]Department of Anesthesiology, University of California, San Diego, San Diego, United States; [4]Medical Physics, University Hospitals Bristol & Weston NHS Trust, Bristol, United Kingdom; [5]Department of Neuroimaging, Institute of Psychiatry, Psychology & Neuroscience, King's College London, London, United Kingdom; [6]Wellcome Wolfson Brain Imaging Centre, School of Psychology, University of East Anglia, Norwich, United Kingdom

*For correspondence:
tony.pickering@bristol.ac.uk

**Summary** Pain perception is decreased by shifting attentional focus away from a threatening event. This attentional analgesia engages parallel descending control pathways from anterior cingulate (ACC) to locus coeruleus, and ACC to periaqueductal grey (PAG) – rostral ventromedial medulla (RVM), indicating possible roles for noradrenergic or opioidergic neuromodulators. To determine which pathway modulates nociceptive activity in humans, we used simultaneous whole brain-spinal cord pharmacological-fMRI (N = 39) across three sessions. Noxious thermal forearm stimulation generated somatotopic-activation of dorsal horn (DH) whose activity correlated with pain report and mirrored attentional pain modulation. Activity in an adjacent cluster reported the interaction between task and noxious stimulus. Effective connectivity analysis revealed that ACC interacts with PAG and RVM to modulate spinal cord activity. Blocking endogenous opioids with Naltrexone impairs attentional analgesia and disrupts RVM-spinal and ACC-PAG connectivity. Noradrenergic augmentation with Reboxetine did not alter attentional analgesia. Cognitive pain modulation involves opioidergic ACC-PAG-RVM descending control which suppresses spinal nociceptive activity.

## Editor's evaluation

This paper will be of great interest to researchers interested in cognitive modulations of sensory processing as well as in the brain mechanisms of pain. It shows that attentional modulations of pain are associated with changes in neural communication between cortical areas, brainstem, and spinal cord which are sensitive to opioidergic but not to noradrenergic modulations. These findings are conclusively supported by state-of-the-art simultaneous pharmacological fMRI of the brain and the spinal cord.

## Introduction

Pain is a fundamental and evolutionarily conserved cognitive construct that is behaviourally prioritised by organisms to protect themselves from harm and facilitate survival. As such, pain perception is sensitive to the context within which potential harm occurs. There are well-recognised top-down influences on pain that can either suppress (e.g. placebo *Wager and Atlas, 2015* or task engagement *Büssing et al., 2010*) or amplify (e.g. catastrophising *Gracely et al., 2004*, hypervigilance *Crombez et al., 2004* or nocebo *Benedetti and Piedimonte, 2019*) its expression. These processes influence both acute and chronic pain and provide a dynamic, moment-by-moment regulation of pain as an organism moves through their environment.

A simple shift in attention away from a noxious stimulus can cause a decrease in pain perception – a phenomenon known as attentional analgesia. This effect can be considered to be a mechanism to enable focus, allowing prioritisation of task performance over pain interruption (*Eccleston and Crombez, 1999*; *Erpelding and Davis, 2013*). This phenomenon is reliably demonstrable in a laboratory setting (*Miron et al., 1989*) and a network of cortical and brainstem structures have been implicated in attentional analgesia (*Bantick et al., 2002*; *Brooks et al., 2017*; *Bushnell et al., 2013*; *Lorenz et al., 2003*; *Petrovic et al., 2002*; *Peyron et al., 2000*; *Sprenger et al., 2012*; *Tracey et al., 2002*; *Valet et al., 2004*).

We have shown that two parallel pathways are implicated in driving brainstem activity related to attentional analgesia (*Brooks et al., 2017*; *Oliva et al., 2021b*). Projections from rostral anterior cingulate cortex (ACC) were found to drive the periaqueductal grey (PAG) and rostral ventromedial medulla (RVM), which animal studies have shown to work in concert using opioidergic mechanisms to regulate spinal nociception (*Fields, 2004*; *Fields and Basbaum, 1978*; *Heinricher et al., 1994*; *Ossipov et al., 2010*). Similarly, a bidirectional connection between ACC and locus coeruleus (LC) was also directly involved in attentional analgesia. As the primary source of cortical noradrenaline, the LC is thought to signal salience of incoming sensory information (*Aston-Jones and Cohen, 2005*; *Sara and Bouret, 2012*), but can also independently modulate spinal nociception (*De Felice et al., 2011*; *Hirschberg et al., 2017*; *Hughes et al., 2015*; *Llorca-Torralba et al., 2016*). Although these animal studies provide a framework for our understanding of descending control mechanisms that are likely to be mediating attentional analgesia, the network interactions between brain, brainstem, and spinal cord and the neurotransmitter systems involved in producing attentional analgesia have yet to be elucidated in humans. In part, this gap in our knowledge is because of the distributed extent of the network spanning the entire neuraxis from forebrain to spinal cord, which has only relatively recently become accessible using simultaneous imaging approaches in humans (*Cohen-Adad et al., 2010*; *Finsterbusch et al., 2013*; *Islam et al., 2019*).

To address this issue, we conducted a double-blind, placebo-controlled, three-arm, cross-over pharmacological-fMRI experiment to investigate attentional analgesia using whole neuraxis imaging and a well validated experimental paradigm. To engage attention, we utilised a rapid serial visual presentation (RSVP) task (*Brooks et al., 2017*; *Oliva et al., 2021b*; *Potter and Levy, 1969*) with individually calibrated task difficulties (easy or hard), which was delivered concurrently with thermal stimulation (low or high), adjusted per subject, to evoke different levels of pain. We took advantage of recent improvements in signal detection (*Duval et al., 2015*) and pulse sequence design to simultaneously capture activity across the brain, brainstem, and spinal cord (i.e. whole CNS) in a single contiguous functional acquisition with slice-specific z-shimming (*Finsterbusch et al., 2012*). To resolve the relative contributions from the opioidergic and noradrenergic systems, subjects received either the opioid antagonist naltrexone (which we predicted would block attentional analgesia), the noradrenaline re-uptake inhibitor reboxetine (which we expected to augment attentional analgesia), or placebo control. By measuring the influence of these drugs on pain perception, BOLD activity and effective connectivity between a priori specified regions known to be involved in attentional analgesia (ACC, PAG, LC, RVM, spinal cord *Brooks et al., 2017*; *Oliva et al., 2021b*; *Sprenger et al., 2012*), we sought to identify the network interactions and neurotransmitter mechanisms mediating this cognitive modulation of pain.

## Results

A total of 39 subjects (mean age 23.7, range [18 - 45] years, 18 females) completed the three fMRI imaging sessions with a 2 × 2 factorial experimental design (RSVP task difficulty: easy or hard, thermal stimulus intensity: low or high, *Figure 1*). A different drug was administered orally before each scan session (naltrexone [50 mg], reboxetine [4 mg], or placebo), which included whole CNS imaging with slice-specific z-shimming (see *Figure 1—figure supplement 1*).

The behavioural signature of attentional analgesia is a task*temperature interaction, driven by a reduction in pain ratings during the high temperature-hard task condition (*Brooks et al., 2017*; *Oliva et al., 2021a*; *Oliva et al., 2021b*). A first level analysis of the pooled pain behavioural data across all experimental sessions showed: a main effect of temperature (F (1,38) = 221, p = 0.0001, *Figure 2—figure supplement 1*) with higher scores under the high temperature conditions; a main effect of task (F (1,38) = 4.9, p = 0.03); and importantly demonstrated the expected task*temperature interaction consistent with attentional pain modulation (F (1, 38) = 10.5, p = 0.0025, *Figure 2—figure supplement 1*).

To assess the impact of the drugs on attentional analgesia, each experimental session was analysed independently (*Figure 2A*). Attentional analgesia was seen in the placebo condition (task*temperature interaction (F (1, 38) = 11.20, p = 0.0019), driven primarily by lower pain scores in the hard|high vs easy|high condition (37.5 ± 19.4 vs 40.4 ± 19.8, mean ± SD, p = 0.001, effect size of –0.55 (Cohen's $D_z$)). Similarly, subjects given Reboxetine showed a task*temperature interaction (F (1, 38) = 9.023, p = 0.0047), again driven by decreased pain scores in the hard|high vs easy|high condition (31.9 ± 15.8 vs 35.6 ± 15.5, p = 0.0034, $D_z$ = −0.42). In contrast, Naltrexone blocked the analgesic effect of attention with no task*temperature interaction (F (1, 38) = 0.4355, p = 0.5133), hard|high (37.4 ± 17.1) vs easy|high (38.3 ± 17.1), $D_z$ = −0.11). Further analysis of the attentional modulation of pain showed that subjects in both the placebo and reboxetine conditions showed a significant decrease in pain score during the hard task that was not evident in the presence of naltrexone (*Figure 2—figure supplement 2*, one sample t-test). We used equivalence analysis (TOST method described by *Lakens, 2017*) to demonstrate that the plausible magnitude of the attentional analgesic effect under naltrexone was smaller than a 6% ( < 2.3 point) reduction in pain score (p = 0.049) confirming it as being smaller than that seen in the presence of placebo or reboxetine. This effect was specific to attentional analgesia as naltrexone had no effect on the calibrated temperature for the high thermal stimulus or the speed of character presentation for the RSVP task (*Figure 2—figure supplement 3*). Behaviourally these findings indicate that the attentional analgesic effect is robust, reproducible between and across subjects and that it involves an opioidergic mechanism.

We also noted a drug*temperature interaction on pain ratings in the first level analysis (F (2, 76) = 3.2, p = 0.04, *Figure 2—figure supplement 1*). Comparing reboxetine versus placebo showed the presence of a drug*temperature interaction (F (1, 38) = 5.060, p = 0.03, *Figure 2*), with lower pain scores in the presence of reboxetine indicating that it was underpinned by an analgesic effect of the noradrenergic reuptake inhibitor (in contrast naltrexone vs placebo showed no drug*temperature interaction).

### Whole CNS fMRI: main effects and interactions

To determine the neural substrates for attentional analgesia and to identify the possible involvement of the noradrenergic and opioidergic systems, we initially defined a search volume in which to focus subsequent detailed fMRI analyses. This was achieved by pooling individually averaged data across the three experimental imaging sessions to estimate main effects and interactions across all levels of the neuraxis.

### Spinal cord

Following registration to the PAM50 spinal cord template (see *Figure 2—video 1*), a cluster of activation representing the positive main effect of temperature was identified in the left dorsal horn (DH), in the C6 spinal segment (*Figure 2B*, assessed using permutation testing with a left C5/C6 mask, p < 0.05, TFCE corrected). This represents activity in a population of spinal neurons that responded more strongly to noxious thermal stimulation. This *Spinal$_{noci}$* cluster was somatotopically localised, given that the thermal stimuli were applied to the left forearm in the C6 dermatome (and its location was also confirmed without masking, *Figure 2—figure supplement 5*). BOLD parameter estimates

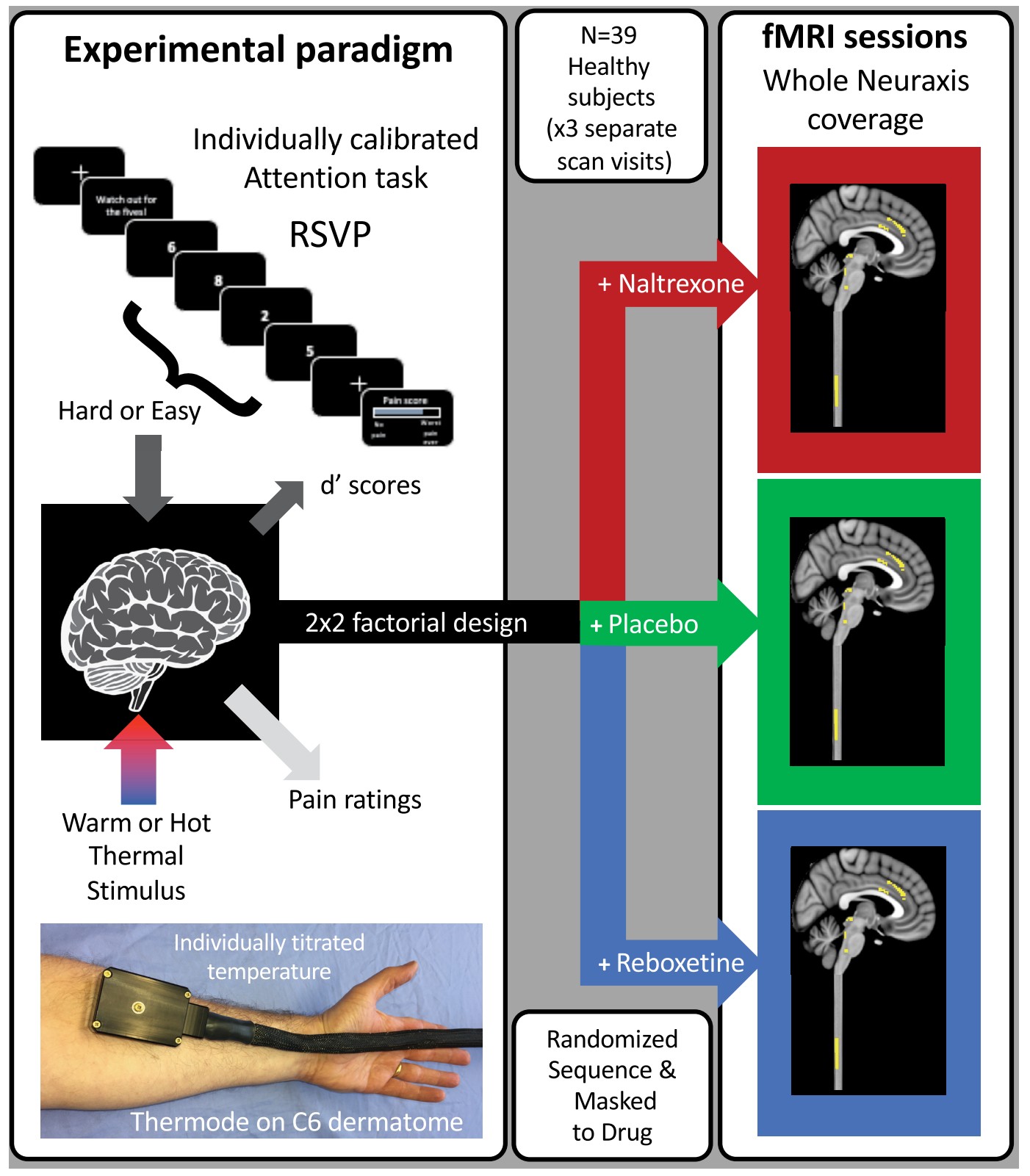

**Figure 1.** Experimental design. A total of 39 healthy subjects had thermal stimulation (to left forearm) while performing a rapid serial visual presentation (RSVP) task. The thermal stimuli were either warm or hot (individually titrated) and the task speed was adjusted for each subject to be either easy or hard (d' 70%, 16 blocks giving four repeats of each condition). This 2 × 2 factorial design allowed the interaction between task and temperature to be tested to identify the attentional analgesic effect. Each subject repeated the experiment on three separate days (at least 1 week apart) with a different drug on

*Figure 1 continued on next page*

*Figure 1 continued*

each occasion (naltrexone, reboxetine, or placebo) and had whole CNS fMRI.

The online version of this article includes the following figure supplement(s) for figure 1:

**Figure supplement 1.** Representative temporal signal to noise ratio (tSNR) data for a single subject, acquired with identical parameters to those used in this study.

were extracted to investigate the activity of this *Spinal$_{noci}$* cluster across the four experimental conditions and three drug sessions (*Figure 2C and D*). There was a positive corelation between the pain ratings and activity in the *Spinal$_{noci}$* cluster across all subjects and experimental conditions (*Figure 2C*). Accordingly in the placebo session, the pattern of BOLD signal change across conditions was similar to the pain scores (*Figure 2A and D*), and the response to a noxious stimulus was lower in the hard|high than easy|high condition, suggesting that the *Spinal$_{noci}$* activity was modulated during attentional analgesia. A similar pattern was evident in the reboxetine condition however, this was not observed in the naltrexone arm consistent with the opioid antagonist-mediated blockade of attentional analgesia. Post hoc analysis of the differences in *Spinal$_{noci}$* BOLD in the hard|high - easy|high conditions, although showing the same pattern of differences in the means, did not show a group level difference between drug sessions. This absence of evidence for attentional modulation of absolute BOLD signal differences may reflect large interindividual differences, low signal to noise in spinal cord fMRI data, or an inability to discriminate between excitatory or inhibitory contributions to measured signal (*Figure 2— figure supplement 2B*).

Analysis of the task*temperature interaction revealed a second discrete spinal cluster (Spinal$_{int}$, *Figure 2B*). This was also located on the left side of the C6 segment but was slightly caudal, deeper and closer to the midline with respect to the *Spinal$_{noci}$* cluster (with only marginal overlap). The location of this activity was again confirmed in an unmasked spinal analysis (*Figure 2—figure supplement 5*). Extraction of BOLD parameter estimates from the *Spinal$_{int}$* cluster in the placebo and reboxetine condition, showed an increased level of activity in the hard|high condition compared to the easy|high and hard|low conditions (*Figure 2E*). The *Spinal$_{int}$* cluster showed significant activation in the hard|high condition in the placebo and reboxetine trials which was not evident in the presence of naltrexone (*Figure 2—figure supplement 2C*). This activity profile suggests this *Spinal$_{int}$* cluster, potentially composed of spinal interneurons, plays a role in the modulation of nociception during the attentional analgesic effect.

## Brainstem and whole brain

To identify the regions of the brainstem involved in mediating attentional analgesia and potentially interacting with the spinal cord, a similar pooled analysis strategy was employed. Activity in brainstem nuclei was investigated using permutation testing with a whole brainstem mask (significant results are reported for $p < 0.05$, TFCE corrected), with subsequent attribution of signal to specific nuclei made through probabilistic masks (from *Brooks et al., 2017*, available from: https://osf.io/xqvb6/). Analysis of the main effect of temperature within a whole brainstem mask showed substantial clusters of activity in the midbrain (PAG) and medulla (RVM) with more discrete clusters in the dorsal pons bilaterally (LC) (*Figure 3A*, *Figure 3—figure supplement 1*). In the main effect of task, the pattern of brainstem activation was more diffuse (*Figure 3B*, *Figure 3—figure supplement 1*), but again included activation of the PAG, RVM, and bilateral LC. Importantly for the mediation of attentional analgesia, no task*temperature interaction was observed within the brainstem (*Figure 3—figure supplement 1*).

Whole-brain analysis of the main effect of temperature (mixed effects analysis, cluster forming threshold $Z > 3.1$, family wise error (FWE) corrected $p < 0.05$) showed activation in pain-associated regions such as primary somatosensory cortex, dorsal posterior insula, operculum, anterior cingulate cortex, and cerebellum with larger clusters contralateral to the side of stimulation (i.e. right side of brain). A cluster in the medial pre-frontal cortex exhibited deactivation. (*Figure 3B and* cluster table in *Figure 3—source data 1*). For the main effect of task, bilateral activation was seen in attention and visual processing areas including lateral occipital cortex, anterior insular cortex, and anterior cingulate cortex. Deactivation was observed in the cerebellum (Crus I), precuneus and lateral occipital cortex (superior division). (*Figure 3B and* cluster table in *Figure 3—source data 1*). No cluster in the whole brain analysis reached significance in the positive task*temperature interaction. Note that cluster

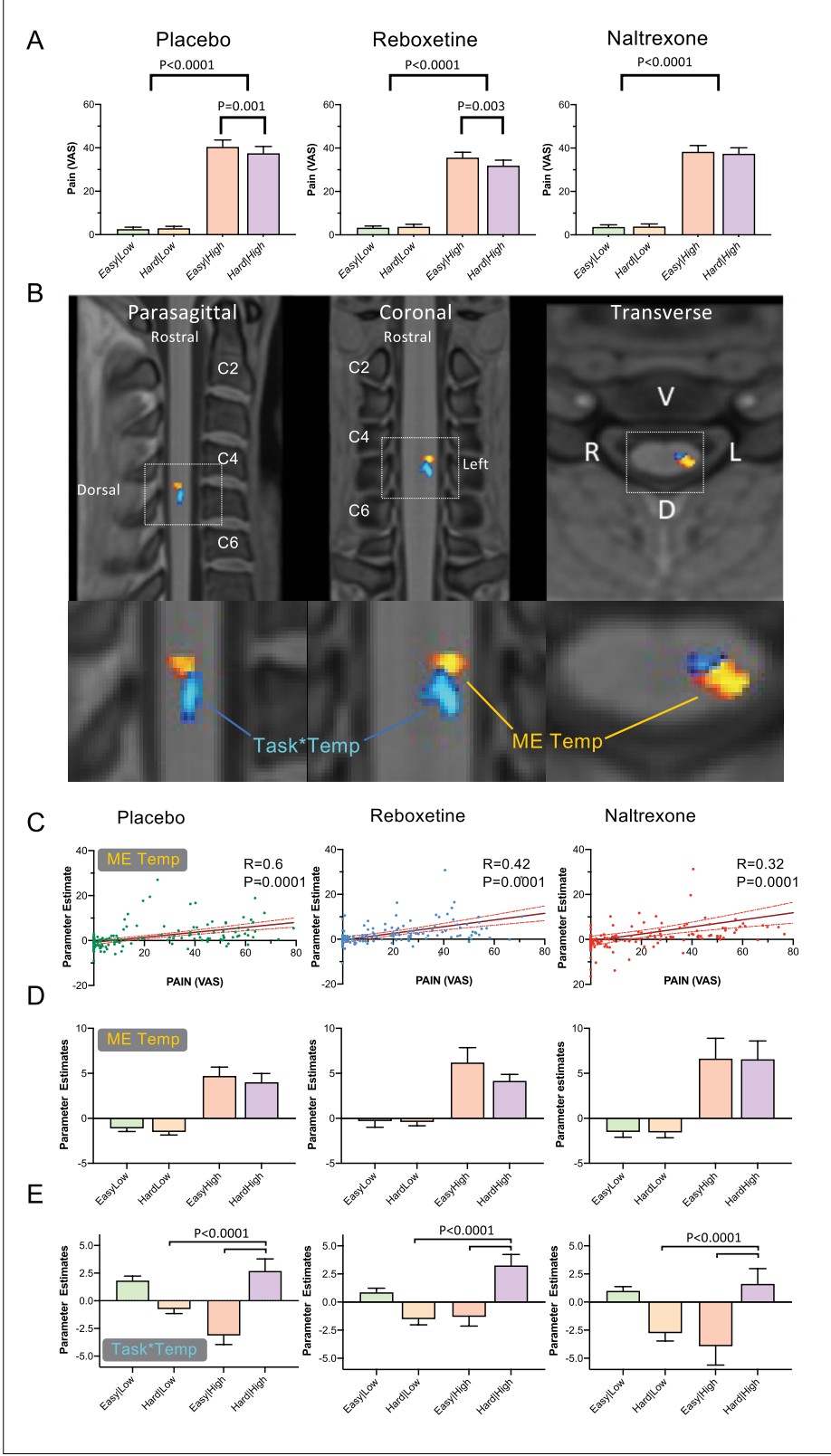

**Figure 2.** Main effect of temperature and task*temperature interaction in the spinal cord. (**A**) Pain scores across the four experimental conditions (i.e. easy|low, hard|low, easy|high and hard|high), for the three drugs. All conditions showed a main effect of temperature (two-way repeated measures ANOVA). Attentional analgesia was seen in the placebo and reboxetine limbs with a task*temperature interaction (F (1, 38) = 11.20, p = 0.0019 and F

*Figure 2 continued on next page*

*Figure 2 continued*

(1, 38) = 9.023, p = 0.004 respectively). In both cases, this was driven by lower pain scores in the hard|high versus easy|high condition (Sidak's post hoc test). In contrast, Naltrexone blocked the analgesic effect of attention as reflected in a loss of the task*temperature interaction (F (1, 38) = 0.4355, p = 0.5133). (**B**) Cervical spine fMRI revealed two distinct clusters of activity within the left side of the C6 cord segment. The first showing the main effect of temperature (red-yellow, $Spinal_{noci}$) and a second showing task*temperature interaction (blue-light blue, $Spinal_{int}$) (significance reported with p < 0.05 (TFCE) within a left sided C5/C6 anatomical mask). No cluster reached significance for the main effect of task. (**C**) Parameter estimates from the $Spinal_{noci}$ cluster showed a positive correlation with the pain scores across all conditions (Pearson's Correlation, 95% CI). (**D**) Parameter estimates from the $Spinal_{noci}$ cluster revealed a decrease in BOLD in the hard|high versus easy|high condition, seen in placebo and reboxetine arms but not in naltrexone. Note the similarity in pattern with the pain scores in (**A**). (**E**) Extraction of parameter estimates from the $Spinal_{int}$ cluster revealed an increase in BOLD in the hard|high condition, across all three drug sessions compared to the easy|high and hard|low conditions (Friedman test p < 0.0001). Mean ± SEM. Parameter estimates extracted from the peak voxel in each cluster.

The online version of this article includes the following video, source data, and figure supplement(s) for figure 2:

**Source data 1.** 2A Pain ratings across contrasts by drug.

**Source data 2.** 2C BOLD parameter estimates from spinal nociception cluster versus pain rating.

**Source data 3.** 2D BOLD parameter estimates for spinal nociception cluster.

**Source data 4.** 2E BOLD parameter estimates for spinal interaction cluster.

**Source data 5.** Pain ratings across conditions by drug.

**Figure supplement 1.** Pain scores under the four experimental conditions (i.e. easy|low, hard|low, easy|high and hard|high), across the three drugs for each of the 39 subjects.

**Figure supplement 1—source data 1.** Figure 2 - figure supplement 1 Second level three way ANOVA of drug versus placebo.

**Figure supplement 2.** Influence of drug on attentional analgesia and on spinal BOLD parameter estimates.

**Figure supplement 2—source data 1.** Figure 2 - figure supplement 2A Difference in pain score between High|Hard and Easy|High conditions by drug.

**Figure supplement 2—source data 2.** Figure 2 - figure supplement 2B Difference in BOLD parameter estimates from spinal nociceptive cluster between High|Hard and Easy|High conditions by drug.

**Figure supplement 2—source data 3.** Figure 2 - figure supplement 2C BOLD parameter estimates from spinal nociceptive cluster in High|Hard condition by drug.

**Figure supplement 3.** Temperature delivered and task speed across the three drug conditions.

**Figure supplement 3—source data 1.** Figure 2 - figure supplement 3 RSVP intercharacter intervals and thermode target temperatures for High thermal stimulus.

**Figure supplement 4.** Analysis of pooled data for main effects and interaction within the cord.

**Figure supplement 5.** Analysis of pooled data for main effects and interaction within the cord.

**Figure 2—video 1.** Registration of functional imaging data to PAM50 template cord.

https://elifesciences.org/articles/71877/figures#fig2video1

---

thresholding does not permit inference on specific voxel locations (*Woo et al., 2014*), we report the full list of regions encompassed by each significant cluster (see *Figure 3* cluster table in *Figure 3—source data 1*).

The distribution of these patterns of regional brain and brainstem activity were closely similar to those found in our previous studies of attentional analgesia (*Brooks et al., 2017*; *Oliva et al., 2021a*; *Oliva et al., 2021b*), with the difference that no area in the brain or brainstem showed a task*temperature interaction (unlike the spinal cord). Parameter maps for all subjects and conditions (in MNI space) for the main effect analyses of brain, brainstem, and spinal cord are available from: https://osf.io/dtpr6/.

## Attentional analgesia and effective network connectivity

Following identification of brain, brainstem, and spinal cord regions active during the attentional analgesia paradigm, and in keeping with our pre-specified regions of interest, we sought to investigate whether their connectivity was altered under the different experimental conditions and whether

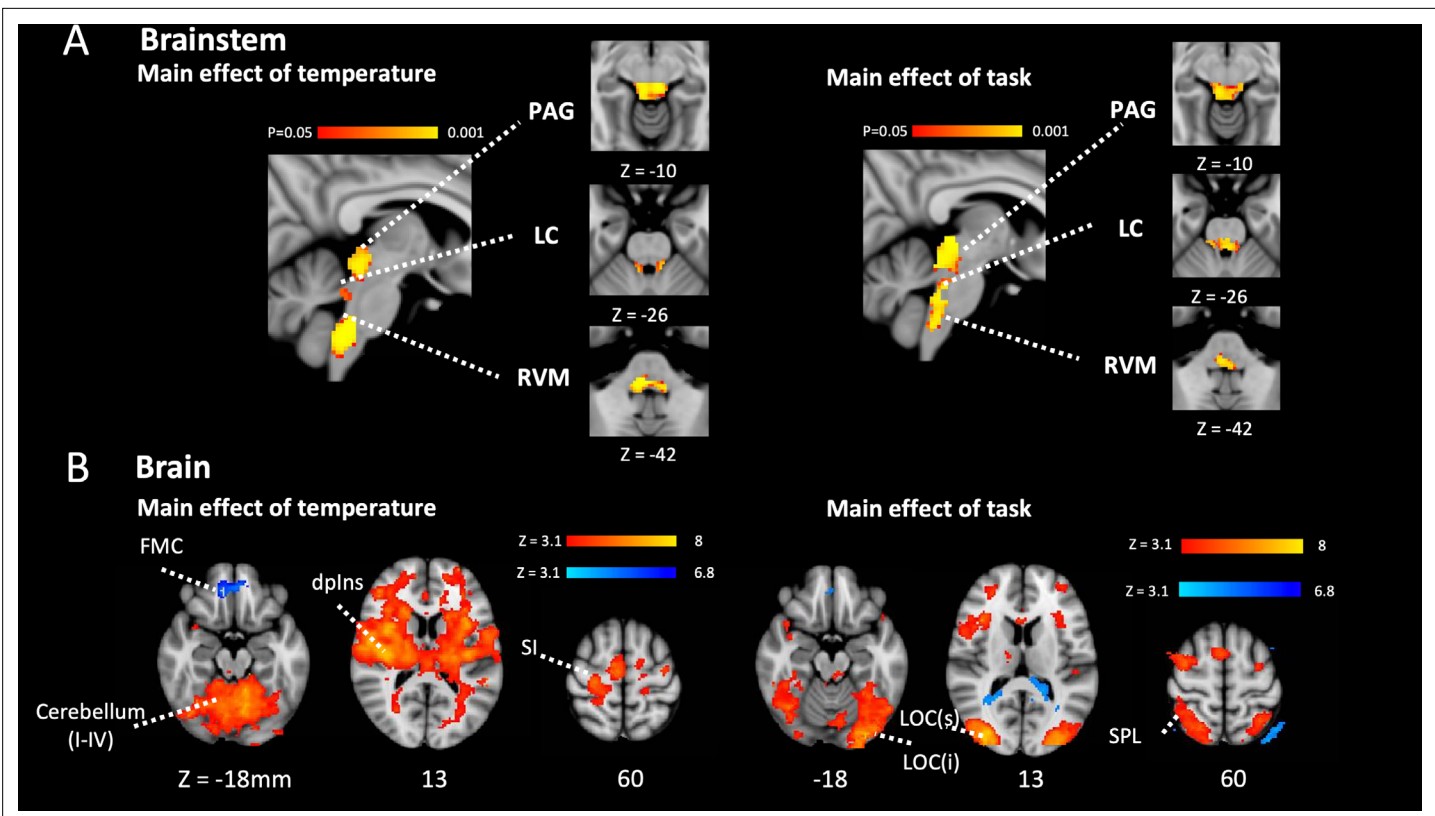

**Figure 3.** Main effect of task and temperature in Brainstem and Cerebrum. (**A**) Main effect of temperature and task in the brainstem after permutation testing with a whole brainstem mask showing clusters of activation in PAG, bilateral LC and RVM. Activity reported with corrected p< 0.05 (TFCE). (**B**) Main effects of temperature and task in brain. In the main effect of temperature contrast there were clusters of activation in a number of pain related sites including in the contralateral primary somatosensory cortex, the dorsal posterior insula and the PAG (red-yellow). The frontal medial cortex deactivated (blue-light blue). In the main effect of task contrast there were clusters of activation in the visual and attention networks including superior parietal cortex, the frontal pole, and the anterior cingulate cortex (red-yellow). The posterior cingulate cortex and lateral occipital cortex showed deactivation (blue-light blue). Activity was estimated with a cluster forming threshold of Z > 3.1 and FWE corrected p < 0.05. (PAG – Periaqueductal grey, LC – Locus coeruleus, RVM – Rostral ventromedial medulla, FMC – Frontomedial cortex, dpIns – dorsal posterior insula, SI – primary somatosensory cortex, LOC – Lateral ocipital cortex (sup and inf), SPL Superior parietal lobule.).

The online version of this article includes the following source data and figure supplement(s) for figure 3:

**Source data 1.** Cluster sizes, peak Z-scores, locations and anatomical locations for each experimental contrast.

**Figure supplement 1.** Whole brain mixed effects analysis of pooled data (inputs are the average of each subject's three sessions) for the three contrasts (main effects of temperature, task and their interaction).

**Figure supplement 2.** Anterior Insula and medulla response after Naltrexone administration.

---

this was subject to specific neurotransmitter modulation. To determine the baseline evidence for the attentional analgesia network, we performed a generalised psychophysiological interaction (gPPI) analysis for the placebo condition alone within the a priori identified seed/target regions (after *Brooks et al., 2017*; *Oliva et al., 2021b*) and based on previous human (*Eippert et al., 2009b*; *Tracey et al., 2002*) and animal studies (*Fields, 2004*; *Ossipov et al., 2010*) of descending control: ACC, PAG, right LC, RVM and cervical spinal cord (left C5/C6 mask).

The gPPI identified the following pairs of connections [seed-target] as being significantly modulated by our experimental conditions (*Figure 4A*, *Figure 4—figure supplements 1 and 2*):

- main effect of temperature [PAG-rLC], [rLC-ACC], [rLC-RVM] and [RVM-spinal cord]
- main effect of task [RVM-rLC] and [PAG-ACC]
- task*temperature interaction [RVM-PAG], [RVM-rLC], and [RVM-spinal cord].

This pattern of network interactions has a number of common features shared with our previous analysis (*Oliva et al., 2021b*) including the task modulation of connectivity between PAG and ACC

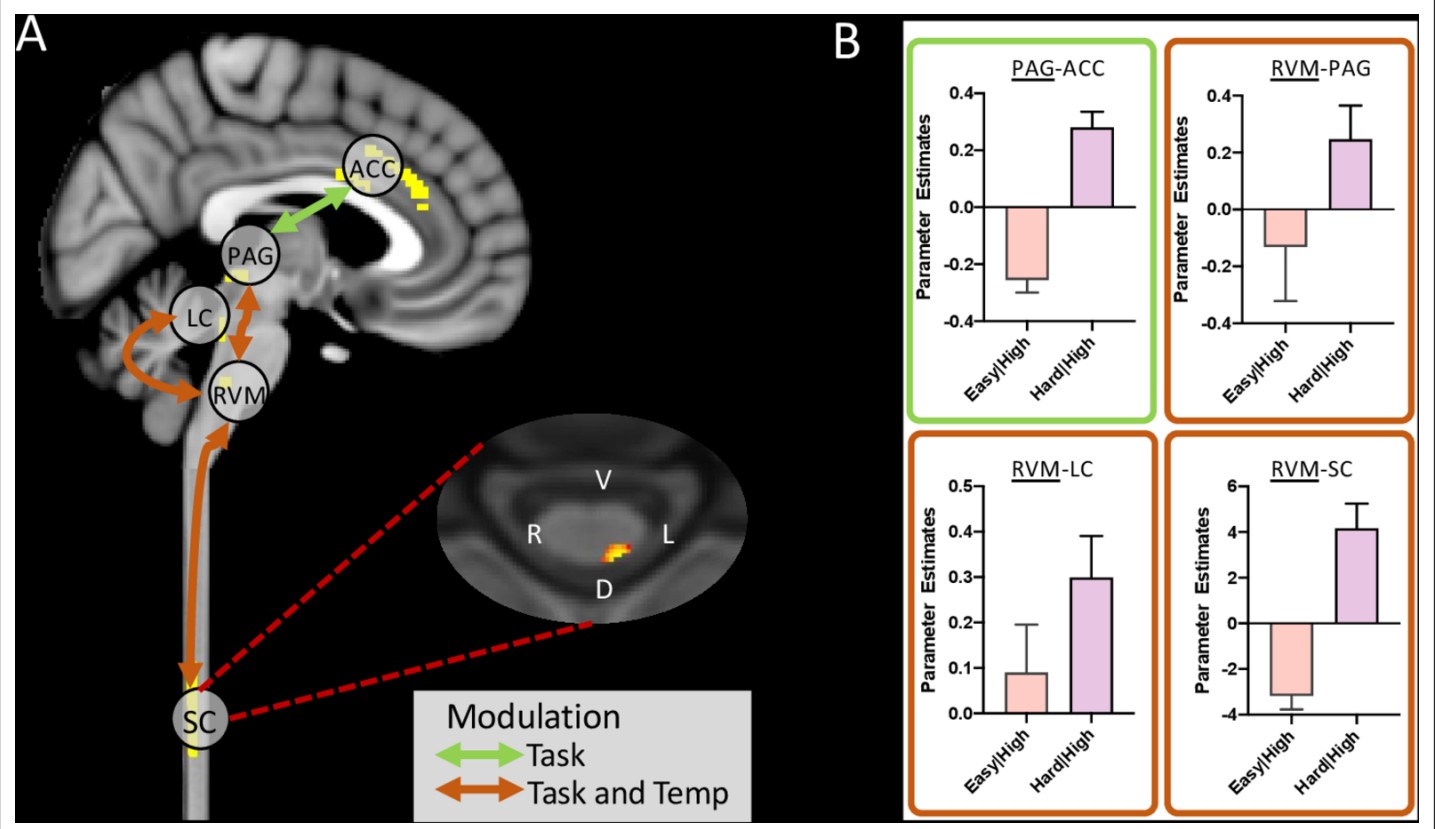

**Figure 4.** Summary of significant connection changes revealed by the gPPI analysis (placebo condition only). (**A**) Permutation testing revealed a significant change in connectivity in the main effect of task contrast between ACC and PAG, and in the task*temperature interaction contrast between PAG and RVM, LC and RVM, and importantly RVM and spinal cord. Masks used for time-series extraction are shown in the sagittal slice (yellow). The spinal cord axial slice shows the voxels with significantly connections with RVM (threshold at = 0.1 for visualisation purposes). (**B**) Extraction of parameter estimates revealed an increase in coupling in the analgesic condition for all of these connections (i.e. hard|high). (Mean ± SEM).

The online version of this article includes the following source data and figure supplement(s) for figure 4:

**Source data 1.** gPPI parameter estimates across connections and conditions.

**Figure supplement 1.** Unmasked whole brain group data for effective connectivity analysis of the placebo condition only.

**Figure supplement 2.** Unmasked group cord data from connectivity analysis of the placebo condition shown on the PAM50 spinal cord template.

and the effect of the task*temperature interaction on connectivity between RVM and PAG. The new features were the important linkage between the spinal cord activity and RVM which is modulated by both temperature and the task*temperature interaction and also the influence of all conditions on communication between RVM and rLC.

Parameter estimates extracted from the connections modulated by task, revealed that the PAG-ACC, RVM-PAG, RVM-rLC, and RVM-Spinal cord connections were stronger in the hard|high versus the easy|high condition (*Figure 4B*), consistent with their potential roles in attentional analgesia.

## Impact of neuromodulators on regional brain activations and network interactions

Having identified this group of regions, in a network spanning the length of the neuraxis, whose activity and connectivity correspond to aspects of the attentional analgesia paradigm we examined whether naltrexone or reboxetine affected the regional BOLD activity or connectivity, comparing each drug against the placebo condition (using paired t-tests).

At the whole brain level, neither drug altered the activations seen for the main effect of temperature. Only the left anterior insula responded more strongly in the presence of Naltrexone for the main effect of task (*Figure 3—figure supplement 2*); however, this was not considered relevant

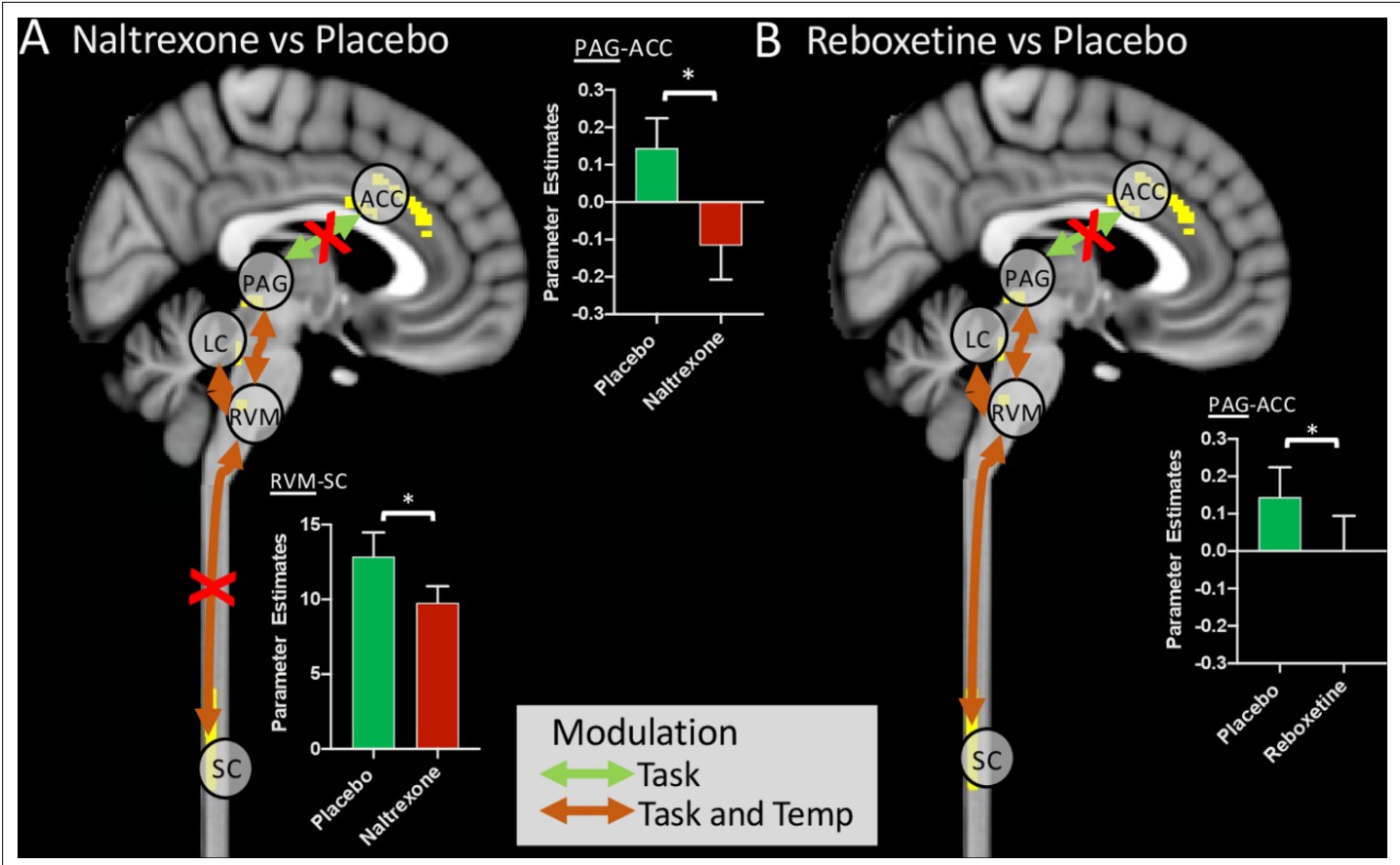

**Figure 5.** Alteration of functional connectivity after dosing with naltrexone or reboxetine compared to placebo. The ACC-PAG connection was significantly weakened by Naltrexone and Reboxetine administration. The RVM-spinal cord connection was significantly weakened by Naltrexone. Red crosses indicate significantly weaker connections after drug. Inset bar plots show BOLD parameter estimates extracted from the PAG-ACC and RVM-spinal cord connections. (Means ± SEM, paired t-test, *p < 0.05).

The online version of this article includes the following source data for figure 5:

**Source data 1.** gPPI parameter estimates for connections by drug.

to the analgesic effect as our behavioural findings showed no effect of naltrexone on task performance (*Figure 2—figure supplement 3B*). In the brainstem, a stronger response to temperature was detected in the lower medulla in the presence of naltrexone compared to placebo (*Figure 3—figure supplement 2*). There was no difference between naltrexone and placebo in the main effect of task in the brainstem. Similarly, no differences in either main effect were uncovered in the brainstem for the reboxetine versus placebo comparison.

The relative lack of effect of either drug on absolute BOLD signal changes provided little evidence for the localisation of their effects in either blocking attentional analgesia (naltrexone) or producing antinociception (reboxetine). However, it has previously been demonstrated that administration of opioidergic antagonists such as naloxone have measurable effects on neural dynamics assessed with fMRI (e.g. *Eippert et al., 2009a*). Therefore, we investigated the network of brain, brainstem and spinal regions that show effective connectivity changes associated with attentional analgesia (under the placebo condition) and explored whether these patterns were altered in the presence of reboxetine or naltrexone (paired t-tests versus placebo).

The administration of naltrexone, which abolished attentional analgesia behaviourally, significantly reduced the connection strength of RVM-spinal cord in the task*temperature interaction (*Figure 5*), indicating a role for opioids in this network interaction. The communication between ACC and PAG was also significantly weakened by both naltrexone and reboxetine, suggesting this connection to be modulated by both endogenous opioids and noradrenaline (*Figure 5*). The strength of the RVM-LC

connection in the main effect of temperature was significantly diminished by reboxetine. None of the other connections in the network were altered significantly by the drugs compared to placebo.

## Discussion

Using brain, brainstem, and spinal cord fMRI we have been able to simultaneously measure the changes in neural activity during this attentional pain modulation study at all levels of the neuraxis during a randomised, placebo-controlled, crossover pharmacological study. This approach allowed unambiguous identification of the nociceptive signal at its site of entry in the dorsal horn and revealed that the task-driven cognitive reductions in pain perception echo the change in absolute BOLD signal at a spinal level. Remarkably the spinal imaging also identified a nearby cluster of neural activity that tracked the interaction between cognitive task and thermal stimulus. Analysis of effective connectivity between brain and brainstem regions and the spinal cord in a single acquisition allowed extension from previous findings (*Brooks et al., 2017*; *Oliva et al., 2021a*; *Oliva et al., 2021b*; *Sprenger et al., 2012*; *Sprenger et al., 2015*) to demonstrate causal changes mediating the interaction of pain and cognitive task including descending influences on the spinal dorsal horn. Naltrexone selectively blocked attentional analgesia and reduced connectivity between RVM and dorsal horn as well as between ACC and PAG. This provides evidence for opioid-dependent mechanisms in the descending pain modulatory pathway that is recruited to mediate the attentional modulation of pain.

The use of individually titrated noxious and innocuous stimuli from a thermode applied to the C6 dermatome of the medial forearm, allowed the identification of a somatotopic $Spinal_{noci}$ cluster in the main effect of temperature contrast in the dorsal horn of the C6 segment. This was strikingly similar to the pattern of activation noted in several previous focussed spinal imaging pain studies in humans (*Brooks et al., 2012*; *Eippert et al., 2009b*; *Sprenger et al., 2012*; *Sprenger et al., 2015*; *Tinnermann et al., 2017*) and non-human primates (*Yang et al., 2015*). The extracted absolute BOLD from the $Spinal_{noci}$ cluster was tightly correlated to the pain scores across the four experimental conditions and therefore the pattern of changes paralleled the changes in pain percept as it was modulated by task. This is similar to the seminal findings from electrophysiological recordings in non-human primates (*Bushnell et al., 1984*), which showed thermal stimulus evoked neural activity in the spinal nucleus of the trigeminal nerve to be altered by attentional focus. Further, it suggested that task related modulation of pain (*Miron et al., 1989*) could occur at the first relay point in the nociceptive transmission pathway. This finding of cognitive modulation of nociceptive input was extended through human spinal fMRI by *Sprenger et al., 2012*, who in a second psychophysical experiment with naloxone provided evidence that the modulation of pain percept may involve opioids. We show that naltrexone attenuates spinal responses to attentional analgesia, which underly the behavioural differences between the high|hard and easy|hard conditions.

Uniquely, our 2 × 2 factorial study design enabled the identification of neural activity reading out the interaction between task and temperature which strikingly was only seen at a spinal level in a cluster located deep and medial to the $Spinal_{noci}$ cluster. The activity in this $Spinal_{int}$ cluster was highest in the high|hard condition (ie when the attentional analgesic effect is seen) and this activation was no longer significant in the presence of naltrexone. This may be consistent with the presence of a local interneuron population in the deeper dorsal horn that could influence the onward transmission of nociceptive information (*Hughes and Todd, 2020*; *Koch et al., 2018*). Such a circuit organisation is predicted by many animal models of pain regulation with the involvement of inhibitory interneurons that shape the incoming signals from the original gate theory of *Melzack and Wall, 1965* through to descending control (*Millan, 2002*). For example, opioids like enkephalin are released from such local spinal inter-neuronal circuits (*Corder et al., 2018*; *François et al., 2017*) and similarly descending noradrenergic projections exert their influence in part via inhibitory interneurons and an alpha1-adrenoceptor mechanism (*Baba et al., 2000a*; *Baba et al., 2000b*; *Gassner et al., 2009*; *Yoshimura and Furue, 2006*). As such the ability to resolve this $Spinal_{int}$ cluster may open a window into how such local interneuron pools are recruited to shape nociceptive transmission in humans according to cognitive context.

Since our goal was to explore the functional connections between brain, brainstem, and spinal cord, we opted to use a single acquisition, with identical imaging parameters (e.g. orientation of slices, voxel dimensions, point spread function) for the entire CNS. This differs from other approaches using different parameters for spinal and brain acquisitions in two fields of view (*Finsterbusch et al.,*

*2012*; *Finsterbusch et al., 2013*; *Islam et al., 2019*; *Sprenger et al., 2015*; *Tinnermann et al., 2017* and reviewed in *Tinnermann et al., 2021*). Our choice was motivated by (i) the need to capture signal across the entire CNS region involved in the task (including the entire medulla), and (ii) that the use of different acquisition parameters for brain and spinal cord could be a confounding factor, particularly for connectivity analyses, due to altered BOLD sensitivity and point-spread function for the separate image acquisitions. By taking advantage the z-shimming approach (*Finsterbusch et al., 2012*) and of the recently developed Spinal Cord Toolbox (*De Leener et al., 2017*), we have been able to detect significant BOLD signal changes in response to experimental manipulations, across the entire CNS.

A key objective of the study was to determine how the information regarding the attentional task demand could be conveyed to the spinal cord. Analysis of regional BOLD signal showed activity in both the main effect of task and of temperature in all three of the key brainstem sites PAG, RVM, and LC with no interaction between task and temperature in the brainstem providing little indication as to which area might be mediating any analgesic effect (in line with previous *Oliva et al., 2021b*). However, an interaction effect was observed on the effective connectivity between RVM and dorsal horn, with coupling highest in the high|hard conditions. The importance of this descending connection to the attentional analgesic effect is emphasised by the effect of naltrexone which blocked both the modulation of RVM-spinal cord connectivity and attentional analgesia a behavioural finding previously noted by *Sprenger et al., 2012*. This fits with the classic model of descending pain modulation that has been developed through decades of animal research (*Fields, 2004*; *Ossipov et al., 2010*) that is engaged in situations of fight or flight and also during appetitive behaviours like feeding and reproduction. Here, we identify that the opioidergic system is also engaged moment by moment, in specific contexts, during a relatively simple cognitive tasks and uncover one of its loci of action in humans.

Analysis of effective connectivity also showed evidence for modulation of pathways from ACC to PAG and PAG to RVM by task and the interaction between task and temperature, respectively (in agreement with *Oliva et al., 2021b*). The communication between ACC and PAG was also disrupted by the opioid antagonist naltrexone. This is similar to the previous finding from studies of placebo analgesia where naloxone was shown to disrupt ACC-PAG communication which was also linked to the mediation of its analgesic effects (*Eippert et al., 2009a*), although behavioural findings of additive analgesia from concurrent placebo and attentional analgesia *Buhle et al., 2012* have been used to argue for distinct pathways of mediation. Activation of the analogous ACC-PAG pathway in rats has recently been shown to produce an analgesic effect mediated via an inhibition of activity at a spinal level indicating that it indeed represents a component of the descending analgesic system (*Drake et al., 2021*). Interestingly, this study also found that this system failed in a chronic neuropathic pain model. This provides evidence for top-down control of spinal nociception during distraction from pain, via the ACC-PAG-RVM-dorsal horn pathway. These findings suggest that the ACC primarily signals the high cognitive load associated with the task to the PAG, that recruits spinally-projecting cells in the RVM. Analgesia could be achieved through disinhibition of spinally-projecting OFF-cells (*Heinricher et al., 1994*; *Lau and Vaughan, 2014*; *Roychowdhury and Fields, 1996*), that inhibit dorsal horn neurons both directly via GABAergic and opioidergic projections to the primary afferents (*Morgan et al., 2008*; *Zhang et al., 2015*) and also indirectly via local inhibitory interneuron pools at a spinal level (*François et al., 2017*) reflected in reduced BOLD signal in the *Spinal*$_{noci}$ cluster and activation of the *Spinal*$_{int}$ pool.

Previous human imaging studies have provided evidence for a role of the locus coeruleus in attentional analgesia (*Brooks et al., 2017*; *Oliva et al., 2021b*). We replicate some of those findings in showing activity in the LC related to both task and thermal stimulus as well as interactions between the LC and RVM that were modulated by the interaction between task and temperature. However, we neither found evidence for an interaction between task and temperature nor for a correlation with analgesic effect in the LC that we reported in our previous studies (*Brooks et al., 2017*; *Oliva et al., 2021b*). We also could not demonstrate altered connectivity between the LC and the spinal cord during the paradigm as we anticipated given its known role in descending pain modulation (*Hickey et al., 2014*; *Hirschberg et al., 2017*; *Llorca-Torralba et al., 2016*; *Millan, 2002*; *Oliva et al., 2021b*; *Ossipov et al., 2010*). It is likely that the brainstem focussed slice prescription used previously is necessary for capturing sufficient signal from the LC, and that extending slice coverage to allow inclusion of the spinal cord compromised signal fidelity in this small brainstem nucleus. The noradrenergic manipulation with reboxetine did show a significant analgesic effect which was independent of task

difficulty. This indicates that this dose of reboxetine is capable of altering baseline gain in the nociceptive system, but has no selective effect on attentional pain modulation. We performed a post hoc Bayesian paired t-test analysis contrasting reboxetine with placebo which showed moderate level of confidence in this null effect on attentional analgesia (Bayes Factor 6.8). Reboxetine also modulated a task-dependent connection between ACC and PAG, although this did not appear to influence task performance and so its behavioural significance is uncertain. In interpreting these findings, one potential explanation is that noradrenaline is not involved in attentional analgesia; however, it could also be because of a ceiling effect where the reuptake inhibitor cannot increase the noradrenaline level any further during the attentional task. In this sense, a noradrenergic antagonist experiment, similar to that used to examine the role of the opioids, would be ideal. However, selective alpha2-antagonists are not used clinically and even experimental agents like Yohimbine have a number of issues that would have confounded this study in that they cause anxiety, excitation and hypertension. Therefore, we conclude that were not able to provide any additional causal evidence to support a role of the LC in attentional analgesia, but this likely reflects a limitation of our approach and lack of good pharmacological tools to resolve the influence of this challenging target.

This combination of simultaneous whole CNS imaging with concurrent thermal stimulation and attentional task in the context of pharmacological manipulation, has enabled the identification of long-range network influences on spinal nociceptive processes and their neurochemistry. An important aspect of this approach is that it has enabled the linkage between a large body of fundamental pain neuroscience that focussed on primary afferent to spinal communication and brainstem interactions (nociception) which can be directly integrated to the findings of whole CNS human imaging. This also offers novel opportunities for translational studies to investigate mechanisms and demonstrate drug target engagement. The finding that it is the effective connectivity of these networks that is of importance in the mediation of the effect of attention and the influence of the opioid antagonist reflects recent observations from large scale studies relating psychological measures to functional connectivity (e.g. *Dubois et al., 2018*). In patient populations, this focus on long range connectivity may help to differentiate between processes leading to augmented nociception and/or altered perception and control (e.g. in fibromyalgia *Oliva et al., 2021a*). Finally, we note that the location of the observed interaction between task and temperature indicates that cognitive tasks are integrated to act at the earliest level in the nociceptive transmission pathway introducing the novel concept of spinal psychology.

## Methods
### Data acquisition
#### Participants

Healthy volunteers were recruited through email and poster advertisement in the University of Bristol and were screened via self-report for their eligibility to participate. Exclusion criteria included any psychiatric disorder (including anxiety/depression), diagnosed chronic pain condition (e.g. fibromyalgia), left handedness, recent use of psychoactive compounds (e.g. recreational drugs or antidepressants) and standard MRI-safety exclusion criteria.

The study was approved by the University of Bristol Faculty of Science Human Research Ethics Committee (reference 23111759828). An initial power analysis was done to determine the sample size using the fmripower software (*Mumford and Nichols, 2008*). Using data from our previous study of attentional analgesia (*Brooks et al., 2017*, main effect of task contrast in the periaqueductal grey matter mask) we designed the study to have an 80% power to detect an effect size of 0.425 (one sample t-test) in the PAG with an alpha of 0.05 requiring a cohort of 40 subjects. Of 57 subjects screened, two were excluded for claustrophobia, three were excluded for regular or recent drug use (including recreational), and five were excluded due to intolerance of the thermal stimulus. This was defined as high pain score ( $\geq$ 8/10) for a temperature that should be non-nociceptive ( < 43 °C). In addition, six participants withdrew from the study as they were unable to attend for the full three visits. One participant had an adverse reaction (nausea) to a study drug (naltrexone) and dropped out of the study. One subject was excluded for being unable to perform the task correctly. Thirty-nine participants completed all three study visits (mean age 23.7, range [18 - 45] years, 18 females).

## Calibration of temperature and task velocity

In the first screening/calibration visit, the participants were briefed on the experiment and gave written informed consent. The participants were familiarised with thermal stimulation by undergoing a modified version of quantitative sensory testing (QST) based on the DFNS protocol (*Rolke et al., 2006*). QST was performed using a Pathway device (MEDOC, Haifa, Israel) with a contact ATS thermode of surface area 9 cm² placed on the subject's left forearm (corresponding to the C6 dermatome). Subsequently, the CHEPS thermode (surface area 5.73 cm²) was used at the same site to deliver a 30-s hot stimulus, to determine the temperature to be used in the experimental visits. Each stimulus consisted of a plateau temperature of 36°C to 45°C, and approximately thirty pseudorandomised 'heat spikes' of 2, 3, or 4 degrees superimposed on the plateau, each lasting less than a second. This temperature profile was used in our previous studies (*Brooks et al., 2017*; *Oliva et al., 2021a*; *Oliva et al., 2021b*) to maintain painful perception, while at the same time avoiding sensitisation and skin damage (*Lautenbacher et al., 1995*). Participants received a range of temperatures between 36°C and 45°C, and were asked to rate the sensation they felt for each stimulus, on a scale from 0 (no pain) to 10 (the worst pain imaginable). The stimulus provoking a pain rating of 6 out of 10 at least three times in a row, was used for the 'high' temperature stimulation in the experiment. If the participant only gave pain scores lower than six to all stimuli, then the maximum programmable plateau temperature of 45 °C was used, but with higher temperature spikes of 3, 4, and 5 degrees above, reaching the highest temperature allowed for safety (50 °C maximum).

The session also included a calibration of the rapid serial visual presentation (RSVP) task (*Potter and Levy, 1969*), where participants were asked to spot the number five among distractor characters. The task was repeated 16 times at different velocities (i.e. different inter-character intervals) in pseudorandom order, ranging from 32 to 256ms. To identify the optimal speed for the hard version of the RSVP task (defined as 70% of each subject's maximum d' score), the d' scores for the different velocities were plotted and the curve fit to a sigmoidal function, using a non-linear least squares fitting routine in Excel (Solver). Once parameterised, the target speed for 70% performance was recorded for subsequent use during the imaging session.

## Imaging sessions

Following the screening/calibration session, participants returned for three imaging sessions, spaced at least a week apart. Participants underwent drug screening (questionnaire) and pregnancy testing. After eating a light snack, they were given either an inert placebo capsule, naltrexone (50 mg). or reboxetine (4 mg) according to a randomised schedule. The dose of the opioid antagonist Naltrexone (50 mg) was as per the British National Formulary (BNF) where it is licensed to prevent relapse in opioid or alcohol dependency. Naltrexone is well absorbed with high oral bioavailability and its levels in the serum peak after 1 hr with a half-life of between 8 and 12 hr (*Verebey et al., 1976*). Reboxetine is used for the treatment of depression, and we used the lowest dose recommended by the BNF (4 mg). It has high oral bioavailability (~95%), serum levels peak at around 2 hr after oral administration and it has a half-life of 12 hr (*Fleishaker, 2000*). Both drugs have previously been used for imaging studies and these formed the basis for our choice of dosing and protocol timings. Oral naltrexone (50 mg) produces 95% blockade of mu opioid receptor binding in the brain (assessed with Carfentanil PET, *Weerts et al., 2008*). Additionally, naltrexone (50 mg) altered network activity in a pharmaco-fMRI study and was well tolerated (*Morris et al., 2018*). Oral reboxetine (4 mg) has been used successfully in human volunteer studies of affective bias with fMRI neuroimaging (*Harmer et al., 2003*; *Miskowiak et al., 2007*). Harmer and colleagues reported an effect of the noradrenergic reuptake inhibitor on emotional processing but no effect on performance of a rapid serial visual presentation task.

All tablets were encased in identical gelatine capsules and dispensed in numbered bottles prepared by the hospital pharmacy (Bristol Royal Infirmary, University Hospitals Bristol and Weston NHS Foundation Trust). Neither the participant nor the investigator knew the identity of the drug which was allocated by a computer-generated randomised schedule. No subject reported being aware of whether they had received active drug or placebo (but the effectiveness of masking was not formally assessed post hoc after dosing).

One hour after drug dosing, calibration of the RSVP task was repeated (to control for any effect on performance). Before scanning, participants received the high thermal stimulus at their pre-determined temperature, which they rated verbally. If the rating was 6 ± 1, the temperature was kept the same,

otherwise it was adjusted accordingly (up or down). Neither reboxetine nor naltrexone caused a significant change in pain perception or task velocity during the calibration, as verified with paired t tests (placebo versus reboxetine and placebo versus naltrexone, see *Figure 2—figure supplement 3*). On average, the plateau temperature used for high temperature stimuli was 43.8°C ± 1.25°C. The median inter-stimulus interval for the hard RSVP task was 48ms, range [32-96].

In the MRI scanner, participants performed the RSVP task at either difficulty level (easy or hard) whilst innocuous (low) or noxious (high) thermal stimuli were delivered concurrently to their left forearm. The four experimental conditions (*easy|high, hard|high, easy|low, hard|low*), were repeated four times each, in a pseudo-random order. The hard version (70% d' performance) of the task and the high (noxious) thermal stimulus were calibrated as described above. In the easy version of the task, the inter-character presentation speed was always set at 192ms, except when a participant's hard task velocity of was equal or slower than 96ms, whereby the easy task was set to 256ms. The low (innocuous) thermal stimulus was always set to be a plateau of 36 °C with spikes of 2, 3, and 4 °C above this baseline. Participants performed the task (identifying targets) and provided pain ratings 10 s after the end of each experimental block on a visual analogue scale (0–100), using a button box (Lumina) held in their dominant (right) hand.

### Acquisition of functional images

Functional images were obtained with a 3T Siemens Skyra MRI scanner, and 64 channel receive-only head and neck coil. After acquisition of localiser images, a sagittal volumetric T1-weighted structural image of brain, brainstem and spinal cord was acquired using the MPRAGE pulse sequence, (TR = 2000ms, TE = 3.72ms, TI = 1000ms, flip angle 9°, field of view (FoV) 320 mm, GRAPPA acceleration factor = 2) and 1.0 mm isotropic resolution. Blood oxygenation level dependent (BOLD) functional data was acquired axially from the top of the brain to the intervertebral disc between the C6 and C7 vertebral bodies, with TR = 3000ms, TE = 39ms, GRAPPA acceleration factor = 2, flip angle 90°, FoV 170 mm, phase encoding direction P>>A, matrix size 96 by 96.

Slices were positioned perpendicular to the long axis of the cord for the C5-C6 spinal segments, whilst still maintaining whole brain coverage, and had an in-plane resolution of 1.77 × 1.77 mm and slice thickness of 4 mm and a 40% gap between slices (increased to 45–50% in taller participants). To determine the optimal shim offset for each slice, calibration scans were acquired cycling through 15 shim offsets. For the caudal 20 slices covering from spinal cord to medulla, manual inspection of images determined the optimal shim offset to be used for each subject (*Finsterbusch et al., 2012*). The remaining supraspinal slices were acquired with the first and higher order shim offsets determined using the scanner's automated routine. The ability of z-shimmed whole CNS imaging to adequately capture BOLD signal was assessed through pilot data examining the temporal signal to noise ratio (tSNR) across cord and brain, see *Figure 1—figure supplement 1*.

During scanning, cardiac and respiratory processes were recorded using a finger pulse oximeter (Nonin 7500) and pneumatic bellows (Lafayette), respectively. These physiological signals and scanner triggers were recorded using an MP150 data acquisition unit (BIOPAC, Goleta, CA), and converted to text files for subsequent use during signal modelling.

## Data analysis

### Analysis of pain scores

Pain scores recorded during the experiment were investigated collectively for the three visits using a three-way ANOVA in Prism version eight for Windows (GraphPad Software, La Jolla, California). Any significant interaction was further investigated with two separate three-way ANOVAs (placebo versus naltrexone and placebo versus reboxetine). Finally, each drug condition was analysed individually with three separate two-way ANOVAs. Two-tailed post-hoc tests were used to further investigate any interactions.

### Pre-processing of functional data and single-subject analysis

Functional data were divided into spinal cord and brain/brainstem, by splitting at the top of the odontoid process (dens) of the 2nd cervical vertebra. The resulting two sets of images underwent separate, optimised, pre-processing pipelines.

Spinal cord data was motion corrected with AFNI 2dImReg (*Cox, 1996*), registering all time points to the temporal mean. Data was smoothed with an in-plane 2D Gaussian smoothing kernel of 2mm x 2mm FWHM, using an in-house generated script. The Spinal Cord Toolbox (SCT, v4.1.1) was then used to create a 25 mm diameter cylindrical mask around the entire cord to crop the functional data. The SCT was also used to segment the cord from the cerebrospinal fluid (CSF) and register functional images to the PAM50 template (*De Leener et al., 2018*). Manual intervention was necessary to ensure accurate cord segmentation on EPI data. The registration pipeline included two steps: (1) registration of each subject's T1-weighted structural scan to the PAM50 T1-weighted template, (2) registration of acquired functional images to PAM50 template (T2*-weighted) using the output from step one as an initial warping. The inverse warping fields generated by this process were also used to transform the PAM50 CSF mask to subject space (*Figure 2—figure supplement 4* and Animation 1). The mask was then used to create a CSF regressor for use during correction for physiological noise during first level FEAT analysis (part of FSL *Jenkinson et al., 2012*).

Brain functional data was pre-processed and analysed in FEAT. Pre-processing included smoothing with a 6 mm Gaussian kernel, and motion correction with MCFLIRT (*Jenkinson et al., 2002*). Functional data was unwarped with a fieldmap using FUGUE (*Jenkinson, 2003*), then co-registered to the subject's T1-weighted structural scan using boundary-based registration (*Greve and Fischl, 2009*). Structural scans were registered to the 2 mm MNI template using a combination of linear (FLIRT, *Jenkinson and Smith, 2001*) and non-linear (FNIRT, *Andersson et al., 2007*) registration with 5 mm warp resolution.

Physiological noise correction was conducted for the brain and spinal cord (*Brooks et al., 2008*; *Harvey et al., 2008*) within FEAT, and as recommended for use in PPI analyses (*Barton et al., 2015*). Cardiac and respiratory phases were determined using a physiological noise model (PNM, part of FSL), and slice specific regressors determined for the entire CNS coverage. Subsequently these regressors (which are 4D images) were split at the level of the odontoid process, to be used separately for brain and spinal cord physiological noise correction. For the brain data the PNM consisted of 32 regressors, with the addition of a CSF regressor for the spinal cord, giving a total of 33 regressors for this region.

All functional images were analysed using a general linear model (GLM) in FEAT with high-pass temporal filtering (cut-off 90 s) and pre-whitening using FILM (*Woolrich et al., 2001*). The model included a regressor for each of the experimental conditions (*easy|high*, *hard|high*, *easy|low*, *hard|low*), plus regressors of no interest (task instructions, rating periods), and their temporal derivatives. Motion parameters and physiological regressors were also included in the model to help explain signal variation due to bulk movement and physiological noise. The experimental regressors of interest were used to build the following planned statistical contrasts: positive and negative main effect of temperature (high temperature conditions versus low temperature conditions and vice versa), positive and negative main effect of task (hard task conditions versus easy task conditions and vice versa), and positive and negative interactions.

Activity within the cerebrum was assessed using conventional whole-brain cluster-based thresholding and mixed-effects modelling, based on recent recommendations (*Eklund et al., 2016*). However, such an approach would not have been appropriate for the small, non-spherical nuclei within the brainstem and laminar arrangement of the spinal cord dorsal horn, which will typically have a larger rostro-caudal extent. Here we chose to use probabilistic anatomical masks (from *Brooks et al., 2017* and available from https://osf.io/xqvb6/ and *De Leener et al., 2017*) to restrict analysis to specific regions, along with permutation testing to assess significance levels with threshold free cluster enhancement (TFCE) (*Smith and Nichols, 2009*).

## Group analysis

We used a conservative approach to investigate differences in CNS activity in main effects and interactions due to administration of reboxetine or naltrexone. All first-level analyses, single group averages and pooled analyses were performed with the experimenter masked to the study visit (i.e. drug session). The pooled analysis separately averaged across sessions the brain, brainstem, and spinal cord activation in the planned contrasts (main effects of temperature, task, and their interaction): individual subjects' data were averaged using a within-subject 'group' model (treating variance between sessions as a random effect), and resultant outputs averaged (across subjects) using a mixed

effects model. This allowed the generation of functional masks, to use for investigation of differences between drug conditions.

Generalised psychophysiological interaction (gPPI) analysis (*McLaren et al., 2012*) was used to assess effective connectivity changes between brain, brainstem, and spinal cord during the attentional analgesia experiment. The list of regions to be investigated were specified a priori on the basis of our previous study (*Oliva et al., 2021b*), and included the ACC, PAG, LC, and RVM – to which was added the left side of the spinal cord at the C5/C6 vertebral level. Following partial unblinding to drug, an initial analysis was performed for the placebo visit. This analysis strategy, which examined connectivity between CNS regions identified in the pooled data and previously (*Brooks et al., 2017*; *Oliva et al., 2021b*), was initially limited to examination of the placebo data and largely replicated our earlier findings (*Oliva et al., 2021b*). By identifying those connections that are normally active during attentional analgesia, we could then test whether they are subject to specific neurotransmitter modulation. This involved partial-unmasking to the remaining two conditions (information on the specific drug used was withheld), so paired t-tests could be performed between the connections of interest. Finally, after the analysis was completed the full unmasking was allowed for the purpose of interpretation of paired differences between conditions.

## Pooled analysis – spinal cord
For each subject, parameter maps estimated for each contrast and each visit (i.e. drug session), were registered to the PAM50 template with SCT. Each contrast was then averaged across visits using a within-subject ordinary least squares (OLS) model using FLAME (part of FSL) from command line. The resulting average contrasts (registered to the PAM50 template) were each concatenated across subjects (i.e. each contrast had 39 samples). These were then investigated with a one-sample t-test in RANDOMISE, using a left C5-6 vertebral mask, based on the probabilistic atlas from the SCT. The choice to use a relatively large vertebral level mask, rather than a more focussed grey matter mask, was based on consideration of (1) the voxel size of our fMRI data compared to the high-resolution data (0.5 mm) used to define probabilistic grey matter masks in SCT, and (2) to allow for inter-subject differences in segmental representation of the stimulation site on the left forearm. It should be noted that by using larger masks we effectively decreased our sensitivity to detect activation, due to the more punitive multiple comparison correction. Results are reported with threshold free cluster enhancement (TFCE) p < 0.05 corrected for multiple comparisons. Significant regions of activation from this pooled analysis were used to generate masks for subsequent comparison between conditions, using paired t-tests.

## Pooled analysis – brainstem
Similar to the spinal cord, for each subject, parameter maps from the brainstem for each planned contrast and visit were averaged with an OLS model in FEAT software. The resulting average was the input to a between-subjects, mixed effects, one-sample t-test in FEAT. Subsequently, group activations for each of the six contrasts were investigated with permutation testing in RANDOMISE, using a probabilistic mask of the brainstem taken from the Harvard-Oxford subcortical atlas (threshold set to p = 0.5). Results are reported with TFCE correction and p < 0.05. Significant regions of activity were binarized and used as a functional mask for the between conditions comparison.

## Pooled analysis – brain
Brain data was averaged and analysed with the same FEAT analyses that were applied in the brainstem. Following within subject averaging, group activity was assessed with a mixed effects two-tailed one sample t-test at the whole-brain level, with results reported for cluster forming threshold of Z > 3.1, and corrected cluster significance of p < 0.05. This produced maps of activity (one per planned contrast) that were then binarized to produce masks that were used in follow up paired t-tests.

## Within subject comparison – paired tests
Paired t tests were performed to resolve potential changes in activity in reboxetine versus placebo and naltrexone versus placebo, separately. Design and contrast files for input in RANDOMISE were built in FEAT. A group file with appropriately defined exchangeability blocks was additionally defined. Permutation testing in RANDOMISE was used to assess group level differences between placebo and

the two drugs, separately for brain, brainstem, and spinal cord. The investigation was restricted to the functional masks derived from the main effect an`alysis for each contrast.

### Effective connectivity analysis (gPPI)

For connectivity analysis, functional data for brain, brainstem, and spinal cord were pre-processed as previously described (*Oliva et al., 2021b*). To restrict analysis to connections typically observed during attentional analgesia, we initially estimated the connection pattern for the placebo session, then within this network tested for differences in the other drug conditions. To achieve this goal, placebo data were first analysed for main effects/interaction with the simple (non-gPPI) analysis to define the pattern of BOLD activity. Subsequently, time series extraction was restricted to anatomical regions/contrasts identified previously (*Oliva et al., 2021b*), and a left sided C5/C6 spinal mask which was used to determine spinal cord activation (derived from the spinal cord toolbox *De Leener et al., 2017*). Physiological time-series were extracted from the voxel of greatest significance identified in the analysis of the placebo session, within the prespecified anatomical regions. In particular, time series were extracted from the peak voxel responding to the main effect of temperature in the RVM and spinal cord, the main effect of task in the ACC, PAG and LC, and the task * temperature interaction in the spinal cord (see *Figure 4—figure supplements 1 and 2*).

For gPPI, physiological time-series were included in a GLM that also included the same regressors present in the first level main effects analysis that is regressors for the experimental conditions and all nuisance regressors (rating period, instruction, PNM, movement parameters). Interaction regressors were then built by multiplying the physiological time-series by each of the experimental regressors, and the planned contrasts constructed (e.g. positive main effect of task). Slice timing correction was not used for this connectivity analysis, as (1) there is no clear recommendation for its use (*Harrison et al., 2017*; *McLaren et al., 2012*; *O'Reilly et al., 2012*), (2) it was omitted in a similar cortico-spinal fMRI study (*Tinnermann et al., 2017*), and (3) to be consistent with our previous study (*Oliva et al., 2021b*). Apart from systematically varying the input physiological timeseries corresponding to different seed regions, models used for estimating connectivity for brain and spinal cord seeds were otherwise identical. Estimates of effective connectivity for the group were obtained with permutation testing with RANDOMISE, using as targets the same ROI masks used for time-series extraction. For example, a gPPI analysis with an RVM seed timeseries (taken from the region responding during the main effect of temperature), examined connectivity to brain/brainstem and spinal cord with PAG, LC, ACC, and left C5-6 vertebral masks. To test whether drug administration altered connectivity during attentional analgesia, the significant connections detected in the placebo session were examined for differences in the other drug conditions i.e. the *same masks* were used for time-series extraction for gPPI analysis of the naltrexone/reboxetine conditions. At the group level, two-tailed paired t-tests were used to detect differences with RANDOMISE (TFCE, $p < 0.05$) between placebo and naltrexone, and between placebo and reboxetine visits, as described above.

## Acknowledgements

The authors thank Aileen Wilson (Lead Research Radiographer, CRiCBristol) for her support in running experiments, and the subjects who kindly agreed to take part. This research was funded in whole, or in part, by the Wellcome Trust [203963/Z/16/Z; and 088373/Z/09 /A] and Medical Research Council [MR/N026969/1]. For the purpose of Open Access, the author has applied a CC BY public copyright licence to any Author Accepted Manuscript version arising from this submission.

## Additional information

### Funding

| Funder | Grant reference number | Author |
| --- | --- | --- |
| Wellcome Trust | 203963/Z/16/Z | Valeria Oliva |
| Wellcome Trust | 088373/Z/09/A | Anthony E Pickering |

| Funder | Grant reference number | Author |
|---|---|---|
| Medical Research Council | MR/N026969/1 | Jonathan CW Brooks |

The funders had no role in study design, data collection and interpretation, or the decision to submit the work for publication.

### Author contributions
Valeria Oliva, Conceptualization, Data curation, Formal analysis, Investigation, Visualization, Writing – original draft, Writing – review and editing; Ron Hartley-Davies, Methodology, Software; Rosalyn Moran, Conceptualization, Writing – review and editing; Anthony E Pickering, Conceptualization, Funding acquisition, Methodology, Project administration, Supervision, Visualization, Writing – original draft, Writing – review and editing; Jonathan CW Brooks, Conceptualization, Data curation, Methodology, Project administration, Supervision, Writing – original draft, Writing – review and editing

### Author ORCIDs
Valeria Oliva http://orcid.org/0000-0001-7849-191X
Rosalyn Moran http://orcid.org/0000-0003-0227-6548
Anthony E Pickering http://orcid.org/0000-0003-0345-0456
Jonathan CW Brooks http://orcid.org/0000-0003-3335-6209

### Ethics
Human subjects: The study was approved by the University of Bristol Faculty of Science Human Research Ethics Committee (reference 23111759828). All participants were given a participant information sheet. In the first screening/calibration visit, the participants were briefed on the experiment and gave written informed consent.

### Decision letter and Author response
Decision letter https://doi.org/10.7554/eLife.71877.sa1
Author response https://doi.org/10.7554/eLife.71877.sa2

## Additional files

### Supplementary files
• Transparent reporting form

### Data availability
Source data is provided for Figure 2 (A, C, D, E, and supplementary 1, 2 and 3) and Figure 4 (B) and Figure 5. Un-thresholded statistical maps have been shared in Open Science Framework and are available at the following link: https://osf.io/dtpr6/ and the brainstem regional masks of PAG, LC, RVM are available from https://osf.io/xqvb6/.

The following dataset was generated:

| Author(s) | Year | Dataset title | Dataset URL | Database and Identifier |
|---|---|---|---|---|
| Oliva V, Pickering T, Brooks J | 2021 | Simultaneous brain, brainstem and spinal cord pharmacological-fMRI reveals endogenous opioid network interactions mediating attentional analgesia | https://doi.org/10.17605/OSF.IO/DTPR6 | Open Science Framework, 10.17605/OSF.IO/DTPR6 |

The following previously published dataset was used:

| Author(s) | Year | Dataset title | Dataset URL | Database and Identifier |
|---|---|---|---|---|
| Oliva V, Pickering T, Brooks J | 2021 | Probabalistic anatomical gray matter masks of Periaqueductal grey, Locus coeruleus and Rostroventromedial medulla | https://doi.org/10.17605/OSF.IO/XQVB6 | Open Science Framework, 10.17605/OSF.IO/XQVB6 |

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
