## [Editor Report]

This paper will be of great interest to researchers interested in cognitive modulations of sensory processing as well as in the brain mechanisms of pain. It shows that attentional modulations of pain are associated with changes in neural communication between cortical areas, brainstem, and spinal cord which are sensitive to opioidergic but not to noradrenergic modulations. These findings are conclusively supported by state-of-the-art simultaneous pharmacological fMRI of the brain and the spinal cord.

---

## [Decision Letter]

**Decision letter after peer review:**

Thank you for submitting your article "Simultaneous Brain, Brainstem and Spinal Cord pharmacological-fMRI reveals endogenous opioid network interactions mediating attentional analgesia" for consideration by *eLife*. Your article has been reviewed by 3 peer reviewers, including Markus Ploner as Reviewing Editor and Reviewer #1, and the evaluation has been overseen by Christian Büchel as the Senior Editor.

Essential revisions:

(1) Negative findings in the noradrenergic condition play an important role. You might therefore consider Bayesian statistics which allow for distinguishing between evidence of absence and absence of evidence.

(2) A crucial part of the reasoning is the correspondence between behavioral and neuroimaging effects. So far, the correspondence is mostly based on similar patterns of modulations on the group level. Extending this correspondence to the individual level might provide further support. For instance, the authors might relate the individual behavioral effects to the individual neural effects. This would substantially strengthen the relationship between behavioral and neural effects.

(3) Please discuss the absence of a main effect of task difficulty with reference to previous studies on attentional effects on pain.

(4) Please report and discuss effect sizes at least for the behavioural results. Furthermore, please provide a statistical test for the pharmacological modulation of attentional analgesia, i.e. a task *temperature* drug interaction.

(5) Please test the parameter estimates extracted from the main effect of temperature for a difference between the attentional conditions in the high-temperature condition.

(6) Please modify or justtfy the use of 'functional localizers' for the connectivity analyses which are built based on the 'Placebo' condition and then used for evaluating connectivity differences between 'Placebo' and each of the drugs.

(7) Please explain (i) how the omission of slice-time correction impact on the obtained results (considering the TR of 3s), (ii) why time-courses were only extracted from one seed voxel (considering how noisy single-voxel time-series are, especially in areas such as the brainstem), (iii) and how the choice of using different contrast for seed-voxel identification in different regions impact / bias the connectivity results.

(8) Please explain and modify the correction for multiple comparisons. The pooled analysis has been used to come up with a set of regions in which the other analyses were carried out. It is not clear whether you performed corrections within each of these masks separately or whether you performed correction across one large mask consisting of all these regions. If you chose the former approach, a correction for the number of regions used would be appropriate.

(9) Please consider using probabilistic masks for your target structures of interest (e.g. PAG or spinal gray matter and spinal segmental level masks), the use of which would clearly bolster confidence in the spatial assignment of the reported results. Please provide the masks as (supplementary) figures. Please clarify already in the main text that the spinal cord analysis was based on the left C5/C6 anatomical mask but not on the whole spinal cord analysis.

(10) Please provide details on the pharmacological challenge. (i) What is the expected onset and duration of the drugs they used and how does this relate to the start and duration of their paradigm? (ii) What kind of side effects can be expected from these drugs and were those assessed in a structured way in their participants and compared across drug-conditions? (iii) Were participants asked about their beliefs regarding which drug they had received on which visit (as this is crucial to ascertain the double-blind nature of the study)? (iv) How does their dosing compare to previous studies using these drugs (and are there data on receptor occupancy)?

(11) Please provide as supplementary data (i) group-average tSNR maps or group-average tSNR values per region of interest (so one can judge the data quality obtainable across such a large imaging volume), (ii) transverse cross-sections of a group-average spinal cord mean EPI image in template space (so one can judge the precision of the employed registration procedures), and (iii) unmasked brainstem and spinal cord activation and connectivity results (so one can judge the success of their physiological noise correction procedure).

(12) Please explain details of the study design. (i) fMRI-based power analysis: the power analysis is based on PAG data and a one-sample t-test, which is different from crucial parts of this manuscript (e.g. spinal data having different signal characteristics and different statistical tests being used, such as ANOVAs). (ii) Reasons for choosing this particular heat stimulation model (30s plateau with random spikes), as it is a rather unusual approach (and also give some more details on the calibration procedure, as it is hard to follow right now) (iii) Did imaging sessions occur at the same time of day for each participant? (iv) Do you indeed only have 4 trials per condition?

(13) Please revise and tone down the interpretation of the results. (i) Neither do the data allow to make the claim that the blockade of RVM-DH connectivity by Naltrexone is indeed the reason for the reduced analgesic effect in that condition nor do their data allow to make a direct link between deep dorsal horn interneuron pools and the observed BOLD effects. (ii) Please cite the relevant work by Tinnermann and colleagues, who did not only report brainstem-spinal coupling during a cognitive manipulation (Tinnermann et al., 2017, Science), but have also written a review on the difficulties of cortico-spinal imaging (Tinnermann et al., 2021, Neuroimage). (iii) Considering that the authors demonstrate the involvement of a system (ACC, PAG, RVM) in attentional analgesia that has also been implicated in placebo analgesia, it might be worth to comment on shared/distinct underlying mechanisms across such forms of pain modulation (see also Buhle et al., 2012, Psychological Science). (iv) Please expand on your statement in the Discussion that optimized cortico-spinal fMRI protocols with tailored acquisition parameters (Finsterbusch et al., 2013, Neuroimage; Islam et al., 2019, Magnetic Resonance in Medicine) could create confounds for functional connectivity analyses?

(14) It was difficult to find detailed quantitative descriptions of the results shown in Figure 1C and 1D. For example, the authors conducted some post-hoc analyses using BOLD estimates extracted from the clusters identified from Figure 1B, and thus they should be able to provide statistical values to make comments like "In the placebo session, the pattern of BOLD signal change across conditions was strikingly similar to the pain scores", or "… showed an increased level of activity in the hard | high condition". By not providing quantitative results, these descriptions become meaningless. The authors even used the word "strikingly" without any statistical evidence, weakening the overall credibility of their findings.

(15) Some detailed information about the fMRI analyses, such as thresholding methods, were given in the Method section only (e.g., the whole-brain analysis in Figure 2B). But given that this journal has the Results section earlier than the Methods section, it would be better to provide some basic information about thresholding in the Results section as well.

(16) The authors used TFCE methods for the spinal cord and brainstem analyses but used a different thresholding method for the whole-brain analysis. The reason should be provided. It seems arbitrary and post-hoc in the current version of the manuscript.

(17) They used cluster-level thresholding for the whole-brain analysis. With the cluster extent thresholding, "researchers can only infer that there is signal somewhere within a significant cluster and cannot make inferences about the statistical significance of specific locations within the cluster" (a quote from the *NeuroImage* paper titled "Cluster-extent based thresholding in fMRI analyses: Pitfalls and recommendations"). Their description of the whole brain results does not seem to recognize this pitfall of the cluster extent thresholding. They should note the caution about the interpretation of their whole-brain results.

(18) Naltrexone is also known to reverse the placebo effects. I wonder if they tested this effect as well. It is unclear in which context they provided the placebo pill to participants.

*Reviewer #1 (Recommendations for the authors):*

– The authors should avoid equalling the absence of a statistically significant difference with the similarity of brain responses, e.g. when reporting the spinal BOLD responses under different conditions.

– Harmonizing the layout of Figure 1 and Supplementary Figure 1 would facilitate comparisons of the figures.

*Reviewer #2 (Recommendations for the authors):*

Size and robustness of effects. I would strongly encourage the authors to report effect sizes at least for their behavioural results. While the reported analgesic effect is highly significant, it seems to be of rather small size (~3-point difference on a 0-100 scale for the placebo condition) and so the behavioural relevance is limited – such a small difference might also place limits on the detectability of underlying neurobiological changes. Furthermore, I was surprised by the lack of a statistical test for the pharmacological modulation of attentional analgesia, i.e. a task *temperature* drug interaction: if the authors want to make the claim that Naltrexone disrupted attentional analgesia (i.e. that it relies on an opioidergic component), it is not enough to show a significant effect in the placebo condition and a non-significant effect in the Naltrexone condition – these two effects would need to be statistically compared. I am not at all trying to suggest that any effect not surviving p <.05 is not of interest, but for the sake of scientific rigour the appropriate tests should be carried out (I am even more inclined to make this point because for the imaging data, the authors do actually carry out such comparisons).

Analysis choices for fMRI data. While I was in general impressed by the rigor of the fMRI data analysis approach (e.g. blinding of researcher, use of permutation-based statistics), there are several questions I have with regard to the appropriateness of the chosen analysis approaches for the fMRI data.

1) Why do the authors not test the parameter estimates extracted from the main effect of temperature for a difference between the attentional conditions in the high-temperature condition? This is a test they carry out in the behavioural data (Figure 1a), but not in the imaging data (Figure 1c), although it would be necessary to establish a reduction of spinal cord BOLD responses under distraction. Currently, they only report this descriptively, without providing statistical support.

2) I am particularly concerned about the use of 'functional localizers' for the connectivity analyses: these are built based on the 'Placebo' condition and then used for evaluating connectivity differences between 'Placebo' and each of the drugs – does this procedure not automatically create a bias toward finding stronger effects in the 'Placebo' condition compared to the other condition, because 'testing' is not independent of 'selection'? Interestingly, all reported results show exactly this pattern, i.e. stronger responses in the 'Placebo' condition.

3) With respect to their (very interesting) connectivity results, I wondered about three further things: i) how does the omission of slice-time correction impact on the obtained results (considering the TR of 3s), ii) why were time-courses only extracted from one seed voxel (considering how noisy single-voxel time-series are, especially in areas such as the brainstem), and iii) how does the choice of using different contrast for seed-voxel identification in different regions impact / bias the connectivity results?

4) How did the authors address the multiple comparison problem? I understood that they used the pooled analysis to come up with a set of regions in which to carry out the other analyses, but I was not able to determine whether they performed corrections within each of these masks separately or whether they performed correction across one large mask consisting of all these regions? If they chose the former approach, a correction for the number of regions used would be appropriate.

Identification of small target regions in brainstem and spinal cord. I apologize in advance if I have missed the relevant information, but I did not see the authors make use of available probabilistic masks for their target structures of interest (e.g. PAG or spinal gray matter and spinal segmental level masks), the use of which would clearly bolster confidence in the spatial assignment of the reported results. Otherwise, how would the authors confidently make the assignment of BOLD responses to their small target regions? This is especially relevant due to some acquisition and analysis parameters not being optimal for these small-scale structures (e.g. overall voxel size of ~1.8x1.8x5.6mm and 6mm smoothing kernel in the brainstem).

Pharmacological challenge. There are several details that are needed in order to judge the success of their double-blind pharmacological challenge design but are currently not mentioned.

1) What is the expected onset and duration of the drugs they used and how does this relate to the start and duration of their paradigm?

2) What kind of side effects can be expected from these drugs and were those assessed in a structured way in their participants and compared across drug-conditions?

3) Were participants asked about their beliefs regarding which drug they had received on which visit (as this is crucial to ascertain the double-blind nature of the study)?

4) How does their dosing compare to previous studies using these drugs (and are there data on receptor occupancy)?

Quality of fMRI data. Considering the immense difficulties in obtaining high-quality fMRI data from the entire CNS simultaneously, it would be reassuring to have some more data at hand to judge the quality of the acquired data-set. Therefore, could the authors provide as supplementary data (1) group-average tSNR maps or group-average tSNR values per region of interest (so one can judge the data quality obtainable across such a large imaging volume), (2) transverse cross-sections of a group-average spinal cord mean EPI image in template space (so one can judge the precision of the employed registration procedures), and (3) unmasked brainstem and spinal cord activation and connectivity results (so one can judge the success of their physiological noise correction procedure)?

Study design. A few points regarding the design of the study could be expanded upon to help the reader understand their choices.

(1) I applaud the authors for carrying out an fMRI-based power analysis, but can currently not follow their rationale: the power analysis is based on PAG data and a one-sample t-test, which is different from crucial parts of this manuscript (e.g. spinal data having different signal characteristics and different statistical tests being used, such as ANOVAs).

(2) Could the authors explain why they chose this particular heat stimulation model (30s plateau with random spikes), as it is a rather unusual approach (and also give some more details on the calibration procedure, as it is hard to follow right now)?

(3) Did imaging sessions occur at the same time of day for each participant?

(4) Do they indeed only have 4 trials per condition or did I misunderstand this part?

Interpretation of results and relation to other work.

(1) While their discussion is in general a pleasure to read and nicely links their own results to a large body of animal work, I think in several instances it might be worth to tone down the interpretation of their results. For example, neither do their data allow to make the claim that the blockade of RVM-DH connectivity by Naltrexone is indeed the reason for the reduced analgesic effect in that condition nor do their data allow to make a direct link between deep dorsal horn interneuron pools and the observed BOLD effects.

(2) I would suggest to cite the relevant work by Tinnermann and colleagues, who did not only report brainstem-spinal coupling during a cognitive manipulation (Tinnermann et al., 2017, Science), but have also written a review on the difficulties of cortico-spinal imaging (Tinnermann et al., 2021, Neuroimage). (2) Considering that the authors demonstrate the involvement of a system (ACC, PAG, RVM) in attentional analgesia that has also been implicated in placebo analgesia, it might be worth to comment on shared/distinct underlying mechanisms across such forms of pain modulation (see also Buhle et al., 2012, Psychological Science).

(3) Could the authors expand on their statement in the Discussion that optimized cortico-spinal fMRI protocols with tailored acquisition parameters (Finsterbusch et al., 2013, Neuroimage; Islam et al., 2019, Magnetic Resonance in Medicine) could create confounds for functional connectivity analyses?

P4: Is it appropriate to speak of 'low pain' regarding the low innocuous temperature employed?

P4: Stating hypotheses would be welcome and helpful for the reader (i.e. what is the direction of effects the authors were expecting for naltrexone and reboxetine and what previous animal/human studies would they base this on).

P10: Considering that the authors have used DCM in their earlier work on very similar brain and brainstem data, what is the reason for not using this effective connectivity approach here?

P10: Why did they choose the right LC and RVM and not opt for a bilateral region – is there any evidence from animal studies for contralateral responses that motivates this choice?

P11: In many imaging studies on descending control, the ACC results are located next to the genu of the corpus callosum – but as the ACC connectivity result seems to be located much more posteriorly here, could the authors comment on this in terms of function of these regions?

P21: If the calibration of the RSVP tasked was repeated at each study-visit, why was it carried out during the initial screening visit?

P22: It might make sense to mention here that all behavioural data analyses were repeated-measures ANOVAs and paired t-tests.

P22: The FWHM of the smoothing kernel for the spinal cord data is almost identical to the in-plane voxel size – could the authors explain this choice (as it is somewhat unusual)?

P23: Were the EPI data directly registered to a target image in template space or was this done via an intermediate step using the high-resolution MPRAGE data?

P25: I really liked their 'Pooled analysis' approach (as it should be unbiased and sensitive), but was not able to follow their 'Pooled analysis' strategy for spinal cord, brainstem and brain, as the authors report different strategies for the different regions. Sometimes they report using two different t-tests (mixed-effects and permutation-based) and also seem to use different thresholding strategies (TFCE vs 'classical' cluster-based). Could they re-phrase this part and use one consistent strategy throughout?

P32: Why do the authors report using a mixed ANOVA, when their experimental paradigm is a within-subject repeated-measures design and does not have a 'group' factor?

*Reviewer #3 (Recommendations for the authors):*

I will point out some issues in more detail below:

1) It was difficult to find detailed quantitative descriptions of the results shown in Figure 1C and 1D. For example, the authors conducted some post-hoc analyses using BOLD estimates extracted from the clusters identified from Figure 1B, and thus they should be able to provide statistical values to make comments like "In the placebo session, the pattern of BOLD signal change across conditions was strikingly similar to the pain scores", or "… showed an increased level of activity in the hard | high condition". By not providing quantitative results, these descriptions become meaningless. The authors even used the word "strikingly" without any statistical evidence, weakening the overall credibility of their findings.

2) Some detailed information about the fMRI analyses, such as thresholding methods, were given in the Method section only (e.g., the whole-brain analysis in Figure 2B). But given that this journal has the Results section earlier than the Methods section, it would be better to provide some basic information about thresholding in the Results section as well.

3) The spinal cord analysis was based on the left C5/C6 anatomical mask, but it was unclear until I see the figure caption or Methods. Only reading the main texts provides the impression that the authors conducted the whole spinal cord analysis.

4) They also used the brainstem mask, which should be provided as a figure.

5) The authors used TFCE methods for the spinal cord and brainstem analyses but used a different thresholding method for the whole-brain analysis. The reason should be provided. It seems arbitrary and post-hoc in the current version of the manuscript.

6) They used cluster-level thresholding for the whole-brain analysis. With the cluster extent thresholding, "researchers can only infer that there is signal somewhere within a significant cluster and cannot make inferences about the statistical significance of specific locations within the cluster" (a quote from the *NeuroImage* paper titled "Cluster-extent based thresholding in fMRI analyses: Pitfalls and recommendations"). Their description of the whole brain results does not seem to recognize this pitfall of the cluster extent thresholding. They should note the caution about the interpretation of their whole-brain results.

7) Naltrexone is also known to reverse the placebo effects. I wonder if they tested this effect as well. It is unclear in which context they provided the placebo pill to participants.

[Editors’ note: further revisions were suggested prior to acceptance, as described below.]

Thank you for resubmitting your work entitled "Simultaneous Brain, Brainstem and Spinal Cord pharmacological-fMRI reveals endogenous opioid network interactions mediating attentional analgesia" for further consideration by *eLife*. Your revised article has been evaluated by Timothy Behrens (Senior Editor) and a Reviewing Editor.

The manuscript has been improved but there are some remaining issues that need to be addressed, as outlined below (see recommendations for the authors by reviewer #2).

*Reviewer #1 (Recommendations for the authors):*

My comments and concerns have been adequately addressed.

*Reviewer #2 (Recommendations for the authors):*

The authors have convincingly addressed most of the points I raised, but there are a few outstanding issues that have not been resolved.

Drug-specific effects on attentional analgesia. The authors aim to show that their pharmacological interventions modulate attentional analgesia and the adequate way to do so would be via a 3-way interaction (i.e. drug *task* temperature) in the chosen ANOVA approach. However, the behavioural data do not provide evidence for such an effect (i.e. the newly-added ANOVA table). I appreciate the novel inclusion of equivalence-testing results (though it is impossible for me to follow the details of this analysis, since it is not described in the methods section), but their main ANOVA approach simply does not show a significant effect of drug on attentional analgesia and this needs to be made clear prominently in the manuscript – at the moment, this lack of an effect is not even mentioned in the main text.

Attentional effects in spinal cord responses. I am at a loss to follow the authors' claim in the abstract that "Noxious thermal forearm stimulation generated somatotopic-activation of dorsal horn (DH) whose activity correlated with pain report and mirrored attentional pain modulation.", as well as their lines 169-177, since the figure they added to their response letter shows exactly the opposite: there was no difference between HH and EH in the nociceptive cluster in the dorsal horn (I would actually find it very instructive for the reader if this figure were added to the supplement next to the same plot for the pain ratings; currently Figure 2 Supplement 2). Similarly, in lines 182-185 the authors make statements that are not based on any statistical inference.

Assessment of group differences in connectivity. I appreciate the authors' response on this point, but it has unfortunately not assuaged my concerns at all. Since the authors use a 'pooled' analysis for the standard GLM analyses, why do they not use this 'pooled' approach for the PPI as well? This would mitigate the risk I have mentioned before, namely that the current approach might introduce a significant amount of bias into their results (e.g. partial 'double-dipping', see Kriegeskorte et al., 2009, Nature Neuroscience).

Spinal cord: choice of mask. I am still not able to follow the authors' rationale as to why they use rather unspecific vertebral level & hemi-cord masks instead of the available probabilistic masks, which represent the current gold-standard: i.e. why do they not limit their search to the segment of interest (C6) and the grey matter therein? I would ask the authors to indicate within the manuscript why they are using a mask that does not allow spatial assignment of BOLD responses with respect to segmental level and grey matter.

Spinal cord: assessment of denoising success. It is unfortunate that the authors are not sharing the asked-for unmasked spinal cord data (but still use a cord mask), as such data would be most helpful for assessing the remaining level of physiological noise as well as the local nature of spinal cord BOLD responses (e.g. draining vein contributions). I am not able to follow the authors' argument that "due to masking steps in the registration pipeline, it was not possible to include tissues outside the spinal cord" – showing a few millimetres of data around the cord / CSF should be possible.

*Reviewer #3 (Recommendations for the authors):*

The authors have addressed my comments and requests satisfactorily.

---

## [Author Response]

Essential revisions:(1) Negative findings in the noradrenergic condition play an important role. You might therefore consider Bayesian statistics which allow for distinguishing between evidence of absence and absence of evidence.

The reviewer is correct, with the Noradrenergic manipulation we still see an attentional analgesic effect whose magnitude appears similar to placebo. However, we do find that the noradrenaline reuptake inhibitor has produced a significant reduction in the pain scores in the high temperature condition indicating that the drug has had an effect in our experiment (providing evidence of efficacy at this dose). We thank the reviewers for the suggestion of using Bayesian statistics to clarify and strengthen our findings. We calculated the difference in pain ratings between the Easy|High and Hard|High conditions and performed a Bayesian paired t test for the comparison of Placebo vs Reboxetine, to obtain a likelihood ratio of 6.8:1 (in favour of the null versus alternative hypothesis). According to commonly used thresholds, this is considered to indicate moderate evidence in support of the null hypothesis. Therefore, we provide evidence in support of no difference between the Placebo and Reboxetine conditions and have added this statement to the discussion (P19 para2). See also the new figure showing the magnitude of the attentional analgesic effect (Figure 2 Supplementary Figure 2) and the response to point 4 below.

Note the contrast between the reboxetine and naltrexone manipulations is biologically challenging to interpret (the drugs have two different mechanisms of action and are likely acting on distinct neural substrates). Accordingly, this comparison was not part of our original hypothesis and so we have not included this analysis (which does not show any statistically significant difference) to avoid confusing the reader.

(2) A crucial part of the reasoning is the correspondence between behavioral and neuroimaging effects. So far, the correspondence is mostly based on similar patterns of modulations on the group level. Extending this correspondence to the individual level might provide further support. For instance, the authors might relate the individual behavioral effects to the individual neural effects. This would substantially strengthen the relationship between behavioral and neural effects.

We thank the reviewers for this important suggestion, and we have examined the correlation between pain report and the activity in the Spinal_noci_ cluster. This shows that there is a strong correlation between pain rating (for the placebo condition) and the extracted BOLD parameter estimate. This substantially strengthens the relationship between behaviour and neural effects at an individual level. We have added the new analysis to Figure 2C.

(3) Please discuss the absence of a main effect of task difficulty with reference to previous studies on attentional effects on pain.

We find a main effect of task in our behavioural results which is reported (p6 para 1). We also find a main effect of task in our brainstem and cerebral imaging findings in terms of regional activations in the visual processing and attention networks, deactivation in the default mode network and brainstem activations in the PAG, LC and RVM (p11 para1 and Figure 3). As the reviewer notes, these findings are in line with our previous studies (Brooks et al., 2017; Oliva et al., 2021b) which have previously used the 2x2 experimental design and identified a main effect of task. Other fMRI studies have shown equivalent attention task related changes in pain percept and regional brain activity (e.g. (Bantick et al., 2002; Peyron et al., 1999; Valet et al., 2004)). We present novel data showing that task modulates the *effective* connectivity in the network (RVM-LC and PAG-ACC p12 para 2) and that this connectivity between PAG and ACC is sensitive to pharmacological intervention. We do not find any main effect of task at a spinal level but crucially were able to demonstrate a task temperature interaction, which we postulate is related to the mediation of the attentional analgesic effect and reflects the influence of inputs from the brainstem and we note there is appropriate connectivity between the RVM and the spinal cord in the task temp contrast.

(4) Please report and discuss effect sizes at least for the behavioural results.

The attentional analgesia effect is typically of the order of 6-10% (Peyron et al., 1999; Bantick 2002; Valet 2004; Tracey 2002). We have seen similar effect for example in the placebo condition with a 7.3% reduction in pain scores corresponding to an effect size (Cohen’s D_z_) of 0.55 which is moderate. These analgesic effect sizes are similar to those we found in our previous studies involving three independent cohorts employing the same RSVP task and thermal stimulus paradigm (effect size of 0.58, n=57 healthy subjects, data from Oliva et al., 2021 Neuroimage). We have undertaken additional analysis and added a plot of the attentional analgesia effects (Figure 2 Supplementary Figure 3) and to the Discussion.

The placebo and reboxetine conditions show a significant reduction in pain scores in the high hard condition ie attentional analgesia (P=0.0016 and 0.0126 respectively vs 0.5119 for naltrexone, one sample t-tests). The corresponding effect sizes are -0.55 for Placebo, -0.42 for Reboxetine vs -0.11 in the presence of Naltrexone (Cohen’s D_z_). Using equivalence testing we can provide confidence limits on the magnitude of attentional analgesia seen in the presence of naltrexone using the TOST approach (Lakens, 2017). Using a threshold of a 6% change in pain scores as being a lower boundary for the magnitude of the reported attentional analgesia effects (-2.3 points on the VAS scale) which is less than that seen in the placebo or reboxetine conditions (and in our previous studies e.g. Oliva et al., 2021 Neuroimage) then we can state that the effect size seen in the presence of naltrexone lies below this threshold (P=0.049). (Now added to the Results section P6 Para2)

Furthermore, please provide a statistical test for the pharmacological modulation of attentional analgesia, i.e. a task temperature drug interaction.

We have provided statistical tests of attentional analgesia which are the individual 2 way ANOVAs which show an interaction between task and temperature that is not seen in the presence of naltrexone (P6, Figure 2). The first level 3 way ANOVA was significant for a main effect of temperature and task and also for the drug temp interaction and the temp task interaction (see Author response table 1) which we explored further with the 2way ANOVAs. This analysis did not show a significant 3 way drug *temp* task interaction, potentially due to the influence of additional factors influencing the behavioural response to the low temperature condition or task performance. We have now reported the full analysis output in Figure 2 Supplementary Figure 2 (see Author response table 1).

**Author response table 1. sa2table1:** 

ANOVA table	F (DFn, DFd)	P value
Drug	F (2, 76) = 2.272	P=0.11
Temperature	F (1, 38) = 221.6	P<0.0001
Task	F (1, 38) = 4.869	P=0.034
Drug x Temperature	F (2, 76) = 3.243	P=0.045
Drug x Task	F (2, 76) = 1.210	P=0.30
Temperature x Task	F (1, 38) = 10.50	P=0.0025
Drug x Temperature x Task	F (2, 76) = 1.579	P=0.21

(5) Please test the parameter estimates extracted from the main effect of temperature for a difference between the attentional conditions in the high-temperature condition.

We tested the parameter estimates for the Spinal_noci_ cluster for differences between the *Hard|High* and *Easy|High* conditions. There were no significant differences between Parameter Estimates across the drug conditions. Also see pain scores vs Spinal_noci_ BOLD shown in Figure 2C and comments in response to point 2 above.

**Author response image 1. sa2fig1:** 

(6) Please modify or justtfy the use of 'functional localizers' for the connectivity analyses which are built based on the 'Placebo' condition and then used for evaluating connectivity differences between 'Placebo' and each of the drugs.

The rationale behind the connectivity analysis was as follows: we wished to test for differences in connectivity between CNS areas found active in the “simple” 2x2 ANOVA of “pooled” group data (see Page 27 Para 1). We reasoned that by restricting our analyses to the subset of active regions, we would reduce complexity and increase interpretability of connectivity findings. To this end we also sought to minimize the number of SEED-TARGET connections examined, by first seeking evidence for modulation of these links in the placebo condition. Subsequently we tested whether connections that are modulated by attentional analgesia are subject to specific neurotransmitter modulation. It is worth noting that the analysis of the placebo condition largely replicated the findings from an independent data set as reported in Oliva et al., 2021 Neuroimage.

We have altered the description of the process, and removed the term “functional localizers” as this does not adequately capture the purpose of this analysis: Methods section (page 27 para 2):

“This analysis strategy, which examined connectivity between CNS regions identified in the pooled data, was initially limited to examination of the placebo data and largely replicated our earlier findings (Oliva et al., 2021 Neuroimage). By identifying those connections that are normally active during attentional analgesia, we could then test whether they are subject to specific neurotransmitter modulation”

(7) Please explain (i) how the omission of slice-time correction impact on the obtained results (considering the TR of 3s),

The long block length (30sec) and TR (3sec) used makes the use of slice-timing correction (STC) unnecessary for modelling the basic 2x2 ANOVA results, however, it is possible that STC would have altered the results from our generalized psychophysiological interaction (gPPI) analyses. Whilst the use of physiological noise modelling is recommended when performing PPI analyses (Barton et al., 2015), there is no information on whether similar benefits may be had with STC (Friston et al., 1997; Gitelman et al., 2003; McLaren et al., 2012; O'Reilly et al., 2012; Parker and Razlighi, 2019). Indeed, FSL’s implementation of physiological noise modelling requires that STC is not performed, as this would break the relationship between the slice-specific physiological regressors (capturing the time of acquisition relative to the cardiac/respiratory cycle) and the data as acquired during scanning. We also note that the complex interaction between motion correction parameters and STC is not well understood (Churchill et al., 2012; Sladky et al., 2011). Therefore, we chose not to use STC in our analyses favouring physiological noise correction, and this appears to be consistent with other reports using PPI analysis (e.g. (Harrison et al., 2017; Ploner et al., 2010; Tinnermann et al., 2017)) and we note that similar conclusions were reached in a recent review of methodology for cortico-spinal imaging (Tinnermann et al., 2021).

(ii) why time-courses were only extracted from one seed voxel (considering how noisy single-voxel time-series are, especially in areas such as the brainstem),

Time-courses were extracted from the voxel that showed the highest Z-score for the contrast of interest. By identifying the voxel that most closely matched the applied model we have chosen an approach that eschews potential SNR improvements that may come from averaging over a larger mask or taking a spherical region of interest, and have thus favoured specificity of response as per our earlier study (Oliva et al., 2021). This approach also allows for anatomical heterogeneity between individuals. Lastly, the reviewer notes that single voxel timeseries are noisy – but we would argue that this “noise” is exactly what the gPPI is attempting to assess, and were it to be truly random in nature, we would be highly unlikely to find consistent patterns of connectivity within our target regions (determined using correction for multiple comparisons).

(iii) and how the choice of using different contrast for seed-voxel identification in different regions impact / bias the connectivity results.

As stated above (point 6), the rationale here was to minimize the number of comparisons performed, to increase the interpretability of findings. To this end, we sought to limit our examination to only those connections found present during attentional analgesia in the placebo condition. By restricting our analyses to only those connections and contrasts, we believe we minimize bias and help reduce the number of comparisons performed. The choice of contrasts for brain and brainstem areas was also consistent with our previous paper (Oliva et al., 2021 Neuroimage), with the purpose of testing whether naltrexone and/or reboxetine altered this network. The rationale for choosing a main effect of attention contrast for ACC, PAG and LC, versus a main effect of temperature contrast in the RVM was to attempt to detect the effect of attention on pain: whether task-responding regions are modulating lower-level temperature-responding regions as would be expected for descending control. In the spinal cord, we used both significant contrasts.

The following was added to the Methods (P29, Para 2):

“Slice timing correction was not used for this connectivity analysis, as (1) there is no clear recommendation for its use (Harrison et al., 2017; McLaren et al., 2012; O'Reilly et al., 2012), (2) it was omitted in a similar cortico-spinal fMRI study (Tinnermann et al., 2017) and (3) to be consistent with our previous studies (Oliva et al., 2021b).”

(8) Please explain and modify the correction for multiple comparisons. The pooled analysis has been used to come up with a set of regions in which the other analyses were carried out. It is not clear whether you performed corrections within each of these masks separately or whether you performed correction across one large mask consisting of all these regions. If you chose the former approach, a correction for the number of regions used would be appropriate.

We previously identified a network of brain regions (ACC, PAG, LC, RVM) involved in attentional analgesia, which included brainstem structures originally identified using probabilistic masks (Brooks et al., 2017). Subsequently we showed that with a larger sample (n=57) it was possible to identify the same pattern of activity within a whole brainstem mask (Oliva et al., 2021b), confirming the original results. However, the brainstem and spinal cord remain difficult to image, suffering from low intrinsic signal to noise, signal drop-out and increased influence from physiological noise (Brooks et al., 2013; Eippert et al., 2017; Tinnermann et al., 2021). Were there no evidence from human brain imaging or experimental animal studies for the involvement of these structures in pain processing, we would agree that a suitably cautious and conservative approach to statistical inference would be necessary. However, not only are these structures known to be involved in pain processing, as has been adequately demonstrated by numerous papers using masked analyses (e.g. Baliki et al., 2010; Geuter et al., 2017; Ploner et al., 2010), but they were specified a priori on the basis of our earlier studies (Brooks et al., 2017; Oliva et al., 2021b) and the existing literature.

In this study, rather than use masks for specific brainstem nuclei we determined activity using a probabilistic *whole brainstem* mask (see Results -> Brainstem and whole brain, page 10, and Methods -> Pooled analysis – brainstem page 28, para 2). Cerebral activation patterns were determined using *whole brain* analysis. Both analyses were performed with correction for multiple comparisons. Subsequently, activity within specific regions was extracted using our previously published brainstem masks (LC, PAG, RVM) and the Harvard-Oxford probabilistic cortical atlas (ACC). Whilst the spinal activity was reported using a probabilistic left C5-C6 mask (PAM50 template), it should be noted that the same pattern of activity is found with a *whole cord* analysis (corrected for multiple comparisons), that matches the expected segmental input to the cord given the stimulation site (see Point 11 below and Figure 2 supplementary figure 4). We hope these data give the reviewers (and the reader) confidence that the masking procedure has not biased results for this region, and that a correction for the number of pre-specified masks is unwarranted.

The planned connectivity analysis between areas identified as active in the pooled data (unmasked or masked), was performed within the a priori specified masks, but rather than present uncorrected statistics (as is frequently done for PPI analyses) we have used permutation testing and TFCE correction (P<0.05) to provide confidence that the connectivity changes between each seed and target region are unlikely to have happened by chance.

(9) Please consider using probabilistic masks for your target structures of interest (e.g. PAG or spinal gray matter and spinal segmental level masks), the use of which would clearly bolster confidence in the spatial assignment of the reported results. Please provide the masks as (supplementary) figures. Please clarify already in the main text that the spinal cord analysis was based on the left C5/C6 anatomical mask but not on the whole spinal cord analysis.

We used the probabilistic brainstem masks developed in our lab, and previously reported (Brooks et al., 2017). These masks were estimated from an independent sample and were based on T2-weighted anatomical imaging and so will not introduce bias into the estimation of activation-changes. The brainstem masks have been previously illustrated in (Brooks et al., 2017; Oliva et al., 2021b), are now outlined in Figure 3 Supplementary Figure 1 and we have clearly signposted their location to the reader (P10 para 1), including making them available through the OSF (link): https://osf.io/xqvb6/.

Similarly, spinal cord data were estimated within masks for the vertebral locations that were determined on the basis of probabilistic analysis of imaging data in the Spinal Cord Toolbox “PAM50” template. We added clarification of the mask used for spinal cord analysis in the main text of the manuscript (Results section, p 7):

“assessed using permutation testing with a left C5/C6 mask, P<0.05, TFCE corrected”.

(10) Please provide details on the pharmacological challenge. (i) What is the expected onset and duration of the drugs they used and how does this relate to the start and duration of their paradigm?

For our paradigm participants had an oral dose of active drug or placebo. One hour later they had a calibration RSVP test and determination of their heat pain threshold (using the previously determined speeds and temperatures) before the fMRI session which lasted from 90 to 150 minutes after dosing across the expected maximum concentration of reboxetine and naltrexone. The rationale for the doses and protocols are explained below.

Naltrexone – dose (50mg) as per the British National Formulary (BNF) – used to prevent relapse in formerly opioid or alcohol dependency. Naltrexone is well absorbed with high oral bioavailability and its levels in the serum peak after 1 hour with a half-life of between 8 and 12 hours (Verebey et al., 1976).

Reboxetine is used for the treatment of depression and has a proven safety record. The study used a single oral dose which is the lowest recommended by the BNF (4mg). It has high oral bioavailability (~95%), serum levels peak at around 2 hours after oral administration and it has a half-life of 12 hours (Fleishaker, 2000).

(ii) What kind of side effects can be expected from these drugs and were those assessed in a structured way in their participants and compared across drug-conditions?

The common and significant side effects lists were provided from the BNF to the participants in the study information sheet. Based on previous studies in volunteers with both active drugs we did not expect a high incidence of side effects and participants were not formally screened for side effects but were asked during the study visit and were given contact details for the study team in the event of any symptoms/illness. Only one participant reported a side effect (nausea after receiving naltrexone – during scanning) and withdrew from the study.

(iii) Were participants asked about their beliefs regarding which drug they had received on which visit (as this is crucial to ascertain the double-blind nature of the study)?

Participants were not formally asked whether they knew which drug they were given in any session. However, the drugs were packaged identically and were formulated in gelatin capsules that were indistinguishable. They were allocated according to a randomized schedule by our hospital pharmacy. Additionally, subjects were not given any information about the anticipated effects of the drugs on pain or its attentional modulation in order to avoid expectation effects. The following statement has been added to the methods (P 23 para 2):

“Neither the participant nor the investigator knew the identity of the drug which was allocated by a computer-generated randomised schedule. No subject reported being aware of whether they had received active drug or placebo (but the effectiveness of blinding was not formally assessed post hoc after dosing).”

(iv) How does their dosing compare to previous studies using these drugs (and are there data on receptor occupancy)?

Both drugs have previously been used for imaging studies and these informed our choice of dosing and protocol timings:

50mg of oral naltrexone produces 95% blockade of mu opioid receptor binding in the brain (assessed with Carfentanil PET, (Weerts et al., 2008)). Additionally, pharmaco-fMRI showed effects of naltrexone on network activity using the same dose and timing for the protocol providing a positive neurobiological signal for target engagement (Morris et al., 2018).

This same oral dose of reboxetine (4mg) and delay before testing has been used successfully in human volunteer studies of affective bias with fMRI neuroimaging (Harmer et al., 2003; Miskowiak et al., 2007). Harmer and colleagues reported an effect of the noradrenergic reuptake inhibitor on emotional processing but no effect on performance of a rapid serial visual presentation task. Another imaging study showed an effect of this does of oral reboxetine (4mg) on fear-induced amygdala activation without any evidence of non-specific effects of reboxetine on brain activity (Onur et al., 2009).

This information has been added to the methods (P22-3)

(11) Please provide as supplementary data (i) group-average tSNR maps or group-average tSNR values per region of interest (so one can judge the data quality obtainable across such a large imaging volume),

It was not possible to derive group average tSNR maps, as no suitable resting state data were acquired in this study. Indeed, we did not consider this necessary as we had already performed several pilot studies to determine the tSNR of our technique, and consistency across the areas imaged. This pilot data (N=3) gave average tSNR values for brain (43.1 ± 3.1, range 39.6–45.4) and cord (18.7 ± 4.2, 14.3–22.7), which compare favourably with published simultaneous imaging results, 31.4 ± 8.6 and 8.6 ± 2.1, respectively (Finsterbusch et al., 2013). This difference in SNR (in the cord) is likely due to difference in voxel size (this study: 12.5mm^3^ vs 5mm^3^, (Finsterbusch et al., 2013)) and it points towards data of sufficiently high tSNR to detect BOLD signal change. Note, the *effective* tSNR of the data will be higher through modelling of physiological noise.

(ii) transverse cross-sections of a group-average spinal cord mean EPI image in template space (so one can judge the precision of the employed registration procedures),

Please see Author response image 2 that shows the quality of our registration procedures. The functional image is an average of all functional images (from the placebo visit) registered to the PAM50 template. It is possible to see that not only is the cord centred within the template, but the spinal discs are also aligned.

and (iii) unmasked brainstem and spinal cord activation and connectivity results (so one can judge the success of their physiological noise correction procedure).

Please see unmasked brainstem activity pattern obtained via whole brain mixed effects analysis of pooled data. (Following that there are the equivalent data for the spinal cord, then the data for the connectivity results (brainstem/cord)). The results reflect a group analysis (N=39) of the average response for each subject (i.e. across the 3 sessions) for the 3 conditions (main effects of temperature, task and their interaction). Slices shown (left to right) midline sagittal, coronal through the PAG, bilateral LC and RVM masks, axial at the level of the midline RVM mask. To allow visualisation of the underlying anatomy, data were thresholded at an uncorrected P-value of 0.05 (i.e. Z>1.65). The location of relevant masks are outlined in white, with labels shown. Note the outline of the brainstem mask derived from the Harvard-Oxford sub-cortical probabilistic atlas, which was thresholded at 50% and used for estimating brainstem activity (rather than the whole brain analysis presented here). Assignment of activity to specific nuclei was based on overlap with probabilistic brainstem nuclei masks (Brooks et al., 2017). Positive Z-scores are shown in Red-Yellow colours, whilst negative ones are in Blue-Lightblue. It can be seen that activity was rarely observed in the 4^th^ ventricle, nor in the aqueduct, indicating that physiological noise was adequately corrected for with the chosen scheme, see (Brooks et al., 2008; Kong et al., 2012) for more details. These data are now added to Figure 3 Supplementary Figure 1.

Unmasked cord EPI data from pooled analysis (across all 3 sessions) for the 3 conditions (main effects of task, temperature and their interaction) are shown on the PAM50 spinal cord template. The uncorrected t-score data from RANDOMISE are shown positive (Red-Yellow) and negative (Blue-Light blue), along with the corrected activity (in green) estimated from these data. Vertebral levels are indicated on sagittal section (left side of image). Due to masking steps in the registration pipeline it was not possible to include tissues outside the cord. Activity for the whole cord (i.e. unmasked) is shown for each contrast in green, with TFCE correction P<0.05. These data are now added to Figure 2 Supplementary Figure 4.

Unmasked whole brain group data for effective connectivity analysis of the placebo condition only. For each subject the seed was extracted for the main effect of temperature (within the pooled simple main effects data) within the RVM. I.e. a functional mask was derived from the group data, masked anatomically then applied to each subject separately to identify their peak voxel time series (the seed). Subsequently, the connectivity profile was estimated for each subject using generalised psychophysiological analysis (gPPI), with separate contrasts between the gPPI regressors for the 3 conditions (main effects of task, temperature and their interaction). To allow visualisation of underlying anatomy, data were thresholded at an uncorrected P-value of 0.05 (i.e. Z>1.65). The location of relevant masks are outlined in white (see labels on previous brainstem figure). Positive Z-scores are shown in Red-Yellow colours, whilst negative ones are in Blue-Light blue. Note that significance of connectivity was assessed by using permutation testing within a priori identified anatomical masks. These data are now added as Figure 4 Supplementary Figure 1.

Unmasked group cord data from connectivity analysis of the placebo condition shown on the PAM50 spinal cord template. For each subject the physiological regressor was extracted from a mask representing the main effect of temperature contrast determined with pooled brainstem data within the RVM. Subsequently, generalised psychophysiological interaction (gPPI) regressors were formed for each of the basic contrasts and contrasts between them created. The data represent uncorrected positive (Red-Yellow) and negative (Blue-Lightblue) t-scores, which are the output from RANDOMISE. Vertebral levels are indicated on sagittal section (left side of image). Due to masking steps in the registration pipeline it was not possible to include tissues outside the cord. To aid interpretation of the patterns of activity, the left C5-C6 vertebral mask is shown (white outline). Significant group activity detected within the mask for each contrast are shown in green, with TFCE corrected P<0.05. These data are now added as Figure 4 Supplementary Figure 2.

(12) Please explain details of the study design. (i) fMRI-based power analysis: the power analysis is based on PAG data and a one-sample t-test, which is different from crucial parts of this manuscript (e.g. spinal data having different signal characteristics and different statistical tests being used, such as ANOVAs).

For the a priori power analysis we used the fMRI data from our previous study using an identical experimental paradigm (Brooks et al., 2017). The region of interest used for this purpose (the PAG), is a crucial region of interest for attentional analgesia, and presents similar (although not identical) noise characteristics to the spinal cord data. Because of the novelty of our acquisition with both spinal and brain/brainstem data for an attentional analgesia paradigm, no better dataset was available at that time to inform the calculation. We also note that our study has been adequately powered to identify not only main effects (in the PAG and elsewhere in the brainstem) but also interactions even at a spinal level, to resolve connectivity and to identify effects of drugs on this connectivity. The power calculation was conducted before the study onset, was accepted by our ethical committee and we propose to share the original protocol document as a supplementary file for the sake of transparency. By sharing our imaging data we will make it possible for others to conduct more accurate power calculations for whole neuraxial imaging.

(ii) Reasons for choosing this particular heat stimulation model (30s plateau with random spikes), as it is a rather unusual approach (and also give some more details on the calibration procedure, as it is hard to follow right now)

The stimulation protocol is based on that proposed by (Lautenbacher et al., 1995), and previously used by (Valet et al., 2004) in a block design imaging experiment. The model attempts to produce a “stable and predictable temporal pattern of tonic pain” (Lautenbacher et al., 1995), by means of a constant background of heat (~42-45°C) on to which are superimposed heat spikes. This heating protocol has been used by us across three separate studies (Brooks et al., 2017; Oliva et al., 2021a; Oliva et al., 2021b), and gives rise to stable pain ratings.

Clarification on the heat stimulation model was added to page 22:

“This temperature profile was used in our previous studies (Brooks et al., 2017; Oliva et al., 2021; Oliva et al. 2021) to maintain pain perception, while at the same time avoiding sensitization and skin damage.”

We added detail on the calibration procedure in our methods section (p. 22), which now reads:

“Participants received a range of thermal stimuli between 36 and 45°C, and were asked to rate the sensation they felt for each stimulus during the whole stimulation period, on a scale from 0 (no pain) to 10 (the worst pain imaginable). The temperature which consistently produced a pain rating of 6 out of 10 at least 3 times in a row, was used for the noxious stimulation in the experiment. If the participant only gave pain scores lower than 6 to all stimuli, then the maximum programmable plateau temperature of 45°C was used, but with higher temperature spikes of 3, 4 and 5 degrees above, reaching the highest temperature allowed for safety (50°C maximum).”

(iii) Did imaging sessions occur at the same time of day for each participant?

The imaging sessions did not happen at the same time of the day. All of the experimental sessions started in a 6-hour window between 9am and 3pm (booked as am or pm sessions, 65% of the scans were done in the pm slot). We tried wherever possible to scan participants at the same time of day and achieved that for 54% of participants (which is >4 fold greater than chance). However, this was a challenging study to get the same participants to attend for scanning on 3 different days over a 3 week period and so we prioritized flexibly finding scan slots that worked for the study participants rather than risk losing subjects who had already been through the protocol. There was no evidence of systematic bias in timings across the drug groups as analysis of the distribution of sessions showed no significant difference (Chi^2^(3.68,2), p=0.16).

(iv) Do you indeed only have 4 trials per condition?

Yes, we had four repetitions of each of the four conditions (*hard|hig*h, *easy|high*, *hard|low*, *easy|low*), each lasting 30 seconds. Please see Figure 1 from Brooks et al., 2017 for clarification. Note, the “control” task depicted in this Figure was omitted in the current experiment.

(13) Please revise and tone down the interpretation of the results. (i) Neither do the data allow to make the claim that the blockade of RVM-DH connectivity by Naltrexone is indeed the reason for the reduced analgesic effect in that condition nor do their data allow to make a direct link between deep dorsal horn interneuron pools and the observed BOLD effects.

These are postulates based on our experimental findings and build in part from the animal literature. We have revised and toned down the language to make it clear that we are suggesting that these are interpretations of the data. We think these are both reasonable hypotheses based on our findings (which we have strengthened with the additional analyses) and that there is nothing factually incorrect with either statement.

(ii) Please cite the relevant work by Tinnermann and colleagues, who did not only report brainstem-spinal coupling during a cognitive manipulation (Tinnermann et al., 2017, Science), but have also written a review on the difficulties of cortico-spinal imaging (Tinnermann et al., 2021, Neuroimage).

We thank the reviewer for their comment and have redressed the oversights in not including these papers. The discussion of implementation of PPI in the Science paper was particularly helpful and is now mentioned in the manuscript (Methods p29), as is the review outlining the challenges of cortico-spinal imaging (Discussion – p16, 17)

(iii) Considering that the authors demonstrate the involvement of a system (ACC, PAG, RVM) in attentional analgesia that has also been implicated in placebo analgesia, it might be worth to comment on shared/distinct underlying mechanisms across such forms of pain modulation (see also Buhle et al., 2012, Psychological Science).

The interesting paper by Buhle and colleagues shows that placebo and attentional analgesia are additive suggesting different mechanisms on the basis of behaviour. In part this is based on the idea that a 3-back task saturates the cognitive capacity of PFC and so that it could not produce the additive placebo analgesia. We have cited Buhle et al., in our discussion as a caveat to the argument that they are acting via similar mechanisms (P18).

(iv) Please expand on your statement in the Discussion that optimized cortico-spinal fMRI protocols with tailored acquisition parameters (Finsterbusch et al., 2013, Neuroimage; Islam et al., 2019, Magnetic Resonance in Medicine) could create confounds for functional connectivity analyses?

We have added the following information to the Discussion (see below), which takes into account the difference in bandwidth, echo time, echo train length and voxel size between the “tailored” brain and spinal cord acquisitions, which we felt might reasonably alter BOLD sensitivity and point spread function for the voxel. Clearly there are advantages to using bespoke acquisitions for each region, but until it has been demonstrated that this does not adversely impact relatively insensitive techniques such as PPI, we thought it prudent to stick with an identical acquisition across our regions of interest.

P17 “Our choice was motivated by (i) the need to capture signal across the entire CNS region involved in the task (including the entire medulla), and (ii) that the use of different acquisition parameters for brain and spinal cord could be a confounding factor, particularly for connectivity analyses, due to altered BOLD sensitivity and point-spread function for the separate image acquisitions.”

(14) It was difficult to find detailed quantitative descriptions of the results shown in Figure 1C and 1D. For example, the authors conducted some post-hoc analyses using BOLD estimates extracted from the clusters identified from Figure 1B, and thus they should be able to provide statistical values to make comments like "In the placebo session, the pattern of BOLD signal change across conditions was strikingly similar to the pain scores", or "… showed an increased level of activity in the hard | high condition". By not providing quantitative results, these descriptions become meaningless. The authors even used the word "strikingly" without any statistical evidence, weakening the overall credibility of their findings.

We have now included the correlation between the pain scores and the BOLD activity in the Spinal_noci_ cluster (see answer to Point 2 and Figure 2C) linking behaviour and the imaging findings statistically and robustly at an individual level. This, to a large extent, accounts for the similar pattern of changes in pain scores and spinal BOLD across the conditions and drugs in the graphs in Figure 2A and D. We have removed the term “striking” as requested as the relationships shown in the graphs are clear to the reader.

(15) Some detailed information about the fMRI analyses, such as thresholding methods, were given in the Method section only (e.g., the whole-brain analysis in Figure 2B). But given that this journal has the Results section earlier than the Methods section, it would be better to provide some basic information about thresholding in the Results section as well.

We thank the reviewer for suggesting this necessary clarification. We added this detail to the Results section as follows:

Spinal cord – p. 6: Activity in the spinal cord was assessed using permutation testing with a left C5/C6 mask based on the probabilistic cord atlas available in the spinal cord toolbox (PAM50 template). Significant results are reported for P<0.05, TFCE corrected.

Brainstem – p. 10: Activity in brainstem nuclei was investigated using permutation testing with a whole brainstem mask which was based on the Harvard-Oxford subcortical atlas available in FSLeyes, thresholded at P=0.5 (i.e. > = 50% probability of being brainstem). Significant results are reported for P<0.05, TFCE corrected.

Whole brain – p. 11: Whole-brain analyses were performed in FEAT without ROI masking. Significant results are reported with cluster forming threshold of Z > 3.1, family wise error (FWE) corrected P < 0.05.

(16) The authors used TFCE methods for the spinal cord and brainstem analyses but used a different thresholding method for the whole-brain analysis. The reason should be provided. It seems arbitrary and post-hoc in the current version of the manuscript.

Activity within the cerebrum was assessed using conventional whole-brain cluster-based thresholding and mixed-effects modelling, based on recent recommendations from (Eklund et al., 2016). Such an approach would not have been appropriate for the small, non-spherical nuclei within the brainstem and laminar arrangement of the spinal cord dorsal horn, which will typically have a larger rostro-caudal extent. Here we chose to use anatomical masks to restrict analysis to specific regions, along with permutation testing to assess significance levels. In the permutation tool chosen, RANDOMISE, the most typically applied method for multiple-comparisons correction within a mask is threshold free cluster enhancement (TFCE), which we have used here.

(17) They used cluster-level thresholding for the whole-brain analysis. With the cluster extent thresholding, "researchers can only infer that there is signal somewhere within a significant cluster and cannot make inferences about the statistical significance of specific locations within the cluster" (a quote from the NeuroImage paper titled "Cluster-extent based thresholding in fMRI analyses: Pitfalls and recommendations"). Their description of the whole brain results does not seem to recognize this pitfall of the cluster extent thresholding. They should note the caution about the interpretation of their whole-brain results.

We thank the reviewer for this note. We are indeed aware of the pitfalls of cluster-level thresholding. We believe we addressed this issue by reporting, for each significant cluster, all the regions involved, according to the Harvard-Oxford probabilistic atlas (please see Supplementary table 1). We now added a note of caution for the interpretation of the results at p. 11:

“Note that cluster thresholding does not permit inference on specific voxel locations (Woo et al., 2014), we report the full list of regions encompassed by each significant cluster (see Figure 3 Supplementary Table 1).”

(18) Naltrexone is also known to reverse the placebo effects. I wonder if they tested this effect as well. It is unclear in which context they provided the placebo pill to participants.

The reviewer is correct that opioid antagonists have been shown in some studies to attenuate placebo analgesic effects. However, these studies manipulate the participant’s expectations to embed the prior belief that the inert treatment is in fact a potent analgesic either by suggestion or by classical conditioning. In our study the placebo control session was used as a contrast to the two active drugs (reboxetine and naltrexone). This allowed the identification of specific pharmacological actions of the two drugs without the effect of placebo – so in effect the placebo effect from the capsule administration has been subtracted from the experiment in this design. Therefore, we cannot answer the reviewer’s question as to whether naltrexone blocks any placebo effects. We have added an additional methodology figure (Figure 1) to emphasise the study design with the aim of clarifying the role of placebo.

Reviewer #2 (Recommendations for the authors):P4: Is it appropriate to speak of 'low pain' regarding the low innocuous temperature employed?

We appreciate the point the reviewer is making here and although our low temperature stimulus did produce non-zero average pain ratings at a group level (and so technically this is a low pain state) nonetheless we meant to imply low versus high temperature conditions and have reworded the sentence as below (P. 4, para 2):

“….with individually calibrated task difficulties (easy or hard), which was delivered concurrently with thermal stimulation (low or high), adjusted per subject, to evoke different levels of pain.”

P4: Stating hypotheses would be welcome and helpful for the reader (i.e. what is the direction of effects the authors were expecting for naltrexone and reboxetine and what previous animal/human studies would they base this on).

We have now stated the expected actions of the drug interventions upon attentional analgesia (P 4, para 2). “To resolve the relative contributions from the opioidergic and noradrenergic systems, subjects received either the opioid antagonist naltrexone (which we predicted would block attentional analgesia), the noradrenaline re-uptake inhibitor reboxetine (which we would propose to augment attentional analgesia), or placebo control.” The animal and human evidence for these actions is summarised earlier in the introduction and in the discussion.

P10: Considering that the authors have used DCM in their earlier work on very similar brain and brainstem data, what is the reason for not using this effective connectivity approach here?

We indeed plan to undertake DCM of this rich dataset but that is for a follow-on paper exploring the direction and valency of the connectivity changes and what this can tell us about the mechanisms of analgesia.

P10: Why did they choose the right LC and RVM and not opt for a bilateral region – is there any evidence from animal studies for contralateral responses that motivates this choice?

The RVM is a midline structure and we did not choose/analyse just the right RVM. However, the reviewer is correct that we chose the right (contralateral) LC a priori as that was a finding from our previous similar studies (Oliva et al., 2021 Neuroimage) and there is animal evidence summarised in that paper for a lateralised response to noxious stimulation in the LC.

P11: In many imaging studies on descending control, the ACC results are located next to the genu of the corpus callosum – but as the ACC connectivity result seems to be located much more posteriorly here, could the authors comment on this in terms of function of these regions?

We thank the reviewer for raising this interesting point. We agree that others have identified signal change in and around the genu of the corpus callosum, but would argue that much of that pertains to specific placebo manipulations. Our subjects were fully informed about the exact nature of this study, no manipulation was performed to alter expectations around any of the drugs received, nor was there any covert alteration in applied stimulus temperature to “enhance” perceived pain relief. Instead, we delivered painful thermal stimulation against the background of a cognitively demanding task (after subjects had received their drugs). We note recent discussions around compartmentalisation of the cingulate (van Heukelum et al., 2020), and we acknowledge that our results pertain to both to MCC (involved in conflict resolution between competing attentional demands, amongst other things) and ACC (nociceptive, affective processing). The location of our “ACC” region is sat on the ACC-MCC border by this definition, and likely reflects a combination of task demand and pain processing.

P21: If the calibration of the RSVP tasked was repeated at each study-visit, why was it carried out during the initial screening visit?

The initial calibration was done to both familiarise the subject with the test procedures and to identify the starting point for future sessions. The recalibration before each subsequent session could be expedited and started from this speed to check performance which was often consistent across testing sessions as indicated in Figure 2 Supplementary Figure 3

P22: The FWHM of the smoothing kernel for the spinal cord data is almost identical to the in-plane voxel size – could the authors explain this choice (as it is somewhat unusual)?

The chosen smoothing kernel reflects a desire to match it to the likely size of activation within the cord, whilst still attempting to improve signal to noise ratio. Given that the cross-sectional area of the cord is of the order ~1cm^2^, we felt that this small amount of smoothing provided a reasonable compromise. It should be noted that even though the voxel size (1.77mm) and FWHM of smoothing kernel (2mm) are similar, the smoothing will include regions beyond the nominal 2mm extent of the kernel.

P23: Were the EPI data directly registered to a target image in template space or was this done via an intermediate step using the high-resolution MPRAGE data?

The author is correct in thinking that the spinal cord EPI data were registered to the PAM50 template via an intermediate step registering each subject’s T1-weighted MPRAGE data, which covered brain and spinal cord, to the PAM50 template. This procedure is similar to the registration process for brain EPI data recommended in FSL. This clarification was now added to the manuscript page 25, para 4 (“The registration pipeline included two steps: (1) registration of each subject’s T1-weighted structural scan to the PAM50 T1-weighted template, (2) registration of acquired functional images to PAM50 template (T2*-weighted) using the output from step 1 as an initial warping.”) and is demonstrated and described in Figure 2 animation 1.

P32: Why do the authors report using a mixed ANOVA, when their experimental paradigm is a within-subject repeated-measures design and does not have a 'group' factor?

The reviewer is correct – this was a 3-way repeated measures ANOVA and now changed in the figure legend.

References

Baliki, M.N., Geha, P.Y., Fields, H.L., and Apkarian, A.V. (2010). Predicting value of pain and analgesia: nucleus accumbens response to noxious stimuli changes in the presence of chronic pain. Neuron 66, 149-160.

Bantick, S.J., Wise, R.G., Ploghaus, A., Clare, S., Smith, S.M., and Tracey, I. (2002). Imaging how attention modulates pain in humans using functional MRI. Brain 125, 310-319.

Barton, M., Marecek, R., Rektor, I., Filip, P., Janousova, E., and Mikl, M. (2015). Sensitivity of PPI analysis to differences in noise reduction strategies. J Neurosci Methods 253, 218-232.

Brooks, J.C., Beckmann, C.F., Miller, K.L., Wise, R.G., Porro, C.A., Tracey, I., and Jenkinson, M. (2008). Physiological noise modelling for spinal functional magnetic resonance imaging studies. Neuroimage 39, 680-692.

Brooks, J.C., Davies, W.E., and Pickering, A.E. (2017). Resolving the Brainstem Contributions to Attentional Analgesia. J Neurosci 37, 2279-2291.

Brooks, J.C., Faull, O.K., Pattinson, K.T., and Jenkinson, M. (2013). Physiological noise in brainstem FMRI. Front Hum Neurosci 7, 623.

Churchill, N.W., Oder, A., Abdi, H., Tam, F., Lee, W., Thomas, C., Ween, J.E., Graham, S.J., and Strother, S.C. (2012). Optimizing preprocessing and analysis pipelines for single-subject fMRI. I. Standard temporal motion and physiological noise correction methods. Hum Brain Mapp 33, 609-627.

Eippert, F., Kong, Y., Jenkinson, M., Tracey, I., and Brooks, J.C.W. (2017). Denoising spinal cord fMRI data: Approaches to acquisition and analysis. Neuroimage 154, 255-266.

Eklund, A., Nichols, T.E., and Knutsson, H. (2016). Cluster failure: Why fMRI inferences for spatial extent have inflated false-positive rates. Proc Natl Acad Sci U S A 113, 7900-7905.

Finsterbusch, J., Eippert, F., and Buchel, C. (2012). Single, slice-specific z-shim gradient pulses improve T2*-weighted imaging of the spinal cord. Neuroimage 59, 2307-2315.

Finsterbusch, J., Sprenger, C., and Buchel, C. (2013). Combined T2*-weighted measurements of the human brain and cervical spinal cord with a dynamic shim update. Neuroimage 79, 153-161.

Fleishaker, J.C. (2000). Clinical pharmacokinetics of reboxetine, a selective norepinephrine reuptake inhibitor for the treatment of patients with depression. Clin Pharmacokinet 39, 413-427.

Friston, K.J., Buechel, C., Fink, G.R., Morris, J., Rolls, E., and Dolan, R.J. (1997). Psychophysiological and modulatory interactions in neuroimaging. Neuroimage 6, 218-229.

Geuter, S., Boll, S., Eippert, F., and Buchel, C. (2017). Functional dissociation of stimulus intensity encoding and predictive coding of pain in the insula. *eLife* 6.

Gitelman, D.R., Penny, W.D., Ashburner, J., and Friston, K.J. (2003). Modeling regional and psychophysiologic interactions in fMRI: the importance of hemodynamic deconvolution. Neuroimage 19, 200-207.

Harmer, C.J., Hill, S.A., Taylor, M.J., Cowen, P.J., and Goodwin, G.M. (2003). Toward a neuropsychological theory of antidepressant drug action: increase in positive emotional bias after potentiation of norepinephrine activity. Am J Psychiatry 160, 990-992.

Harrison, T.M., McLaren, D.G., Moody, T.D., Feusner, J.D., and Bookheimer, S.Y. (2017). Generalized Psychophysiological Interaction (PPI) Analysis of Memory Related Connectivity in Individuals at Genetic Risk for Alzheimer's Disease. J Vis Exp.

Kong, Y., Jenkinson, M., Andersson, J., Tracey, I., and Brooks, J.C. (2012). Assessment of physiological noise modelling methods for functional imaging of the spinal cord. Neuroimage 60, 1538-1549.

Lakens, D. (2017). Equivalence Tests: A Practical Primer for t Tests, Correlations, and Meta-Analyses. Soc Psychol Personal Sci 8, 355-362.

Lautenbacher, S., Roscher, S., and Strian, F. (1995). Tonic pain evoked by pulsating heat: temporal summation mechanisms and perceptual qualities. Somatosens Mot Res 12, 59-70.

McLaren, D.G., Ries, M.L., Xu, G., and Johnson, S.C. (2012). A generalized form of context-dependent psychophysiological interactions (gPPI): a comparison to standard approaches. Neuroimage 61, 1277-1286.

Miskowiak, K., Papadatou-Pastou, M., Cowen, P.J., Goodwin, G.M., Norbury, R., and Harmer, C.J. (2007). Single dose antidepressant administration modulates the neural processing of self-referent personality trait words. Neuroimage 37, 904-911.

Morris, L.S., Baek, K., Tait, R., Elliott, R., Ersche, K.D., Flechais, R., McGonigle, J., Murphy, A., Nestor, L.J., Orban, C.*, et al.* (2018). Naltrexone ameliorates functional network abnormalities in alcohol-dependent individuals. Addict Biol 23, 425-436.

O'Reilly, J.X., Woolrich, M.W., Behrens, T.E., Smith, S.M., and Johansen-Berg, H. (2012). Tools of the trade: psychophysiological interactions and functional connectivity. Soc Cogn Affect Neurosci 7, 604-609.

Oliva, V., Gregory, R., Brooks, J.C.W., and Pickering, A.E. (2021a). Central pain modulatory mechanisms of attentional analgesia are preserved in fibromyalgia. Pain in Press.

Oliva, V., Gregory, R., Davies, W.E., Harrison, L., Moran, R., Pickering, A.E., and Brooks, J.C.W. (2021b). Parallel cortical-brainstem pathways to attentional analgesia. Neuroimage 226, 117548.

Onur, O.A., Walter, H., Schlaepfer, T.E., Rehme, A.K., Schmidt, C., Keysers, C., Maier, W., and Hurlemann, R. (2009). Noradrenergic enhancement of amygdala responses to fear. Soc Cogn Affect Neurosci 4, 119-126.

Parker, D.B., and Razlighi, Q.R. (2019). The Benefit of Slice Timing Correction in Common fMRI Preprocessing Pipelines. Front Neurosci 13, 821.

Peyron, R., Garcia-Larrea, L., Gregoire, M.C., Costes, N., Convers, P., Lavenne, F., Mauguiere, F., Michel, D., and Laurent, B. (1999). Haemodynamic brain responses to acute pain in humans: sensory and attentional networks. Brain 122 ( Pt 9), 1765-1780.

Ploner, M., Lee, M.C., Wiech, K., Bingel, U., and Tracey, I. (2010). Prestimulus functional connectivity determines pain perception in humans. Proc Natl Acad Sci U S A 107, 355-360.

Sladky, R., Friston, K.J., Trostl, J., Cunnington, R., Moser, E., and Windischberger, C. (2011). Slice-timing effects and their correction in functional MRI. Neuroimage 58, 588-594.

Tinnermann, A., Buchel, C., and Cohen-Adad, J. (2021). Cortico-spinal imaging to study pain. Neuroimage 224, 117439.

Tinnermann, A., Geuter, S., Sprenger, C., Finsterbusch, J., and Buchel, C. (2017). Interactions between brain and spinal cord mediate value effects in nocebo hyperalgesia. Science 358, 105-108.

Valet, M., Sprenger, T., Boecker, H., Willoch, F., Rummeny, E., Conrad, B., Erhard, P., and Tolle, T.R. (2004). Distraction modulates connectivity of the cingulo-frontal cortex and the midbrain during pain--an fMRI analysis. Pain 109, 399-408.

van Heukelum, S., Mars, R.B., Guthrie, M., Buitelaar, J.K., Beckmann, C.F., Tiesinga, P.H.E., Vogt, B.A., Glennon, J.C., and Havenith, M.N. (2020). Where is Cingulate Cortex? A Cross-Species View. Trends Neurosci 43, 285-299.

Verebey, K., Volavka, J., Mule, S.J., and Resnick, R.B. (1976). Naltrexone: disposition, metabolism, and effects after acute and chronic dosing. Clin Pharmacol Ther 20, 315-328.

Weerts, E.M., Kim, Y.K., Wand, G.S., Dannals, R.F., Lee, J.S., Frost, J.J., and McCaul, M.E. (2008). Differences in δ- and mu-opioid receptor blockade measured by positron emission tomography in naltrexone-treated recently abstinent alcohol-dependent subjects. Neuropsychopharmacology 33, 653-665.

Woo, C.W., Krishnan, A., and Wager, T.D. (2014). Cluster-extent based thresholding in fMRI analyses: pitfalls and recommendations. Neuroimage 91, 412-419.

[Editors' note: further revisions were suggested prior to acceptance, as described below.]

Reviewer #2 (Recommendations for the authors):The authors have convincingly addressed most of the points I raised, but there are a few outstanding issues that have not been resolved.Drug-specific effects on attentional analgesia. The authors aim to show that their pharmacological interventions modulate attentional analgesia and the adequate way to do so would be via a 3-way interaction (i.e. drug task temperature) in the chosen ANOVA approach. However, the behavioural data do not provide evidence for such an effect (i.e. the newly-added ANOVA table). I appreciate the novel inclusion of equivalence-testing results (though it is impossible for me to follow the details of this analysis, since it is not described in the methods section), but their main ANOVA approach simply does not show a significant effect of drug on attentional analgesia and this needs to be made clear prominently in the manuscript – at the moment, this lack of an effect is not even mentioned in the main text.

The 3-way repeated measures ANOVA tests seven null hypotheses relating to main effects and their interactions (we showed significant findings for *Temperature*, *Task*, *TempTask* and *Drug Temp*). In this case the absence of a *Drug* Task *Temp* interaction does not mean that there is no effect of drug on attentional analgesia; any more than its presence would have indicated with any certainty that a drug modulation of attentional analgesic effect was present. Hypothetically, such a 3-way interaction could be caused by other drug effects for example an analgesic action associated with a degree of sedation (such as would be produced by an opioid or alpha2-adrenoceptor agonist) that would have influenced pain scores and task performance differentially (but not have had anything to do with attentional analgesia).

We are not particularly surprised by the absence of a significant 3-way interaction given that attentional analgesia (~7% change in pain scores) is only seen in the difference in pain between the high temperature conditions and that only one of the drugs (naltrexone) selectively influenced attentional analgesia (only evident in the *high|hard* condition ie 1/12^th^ of the observations). In contrast the effect of temperature has a very large influence on the variance in the pain scores and accordingly it dominates the statistics. Nonetheless the presence of significant interactions of *Drug Temp* and *Temp Task* is further investigated in the individual 2-way ANOVA analyses that show the behavioural signature of attentional analgesia is present under the placebo and reboxetine conditions but is attenuated by naltrexone. This is reinforced through equivalence testing, described in the paper of Lakens (2017) and as cited in the text in the relevant Results section (p5 final para).

We also emphasise that our findings are the product of a carefully controlled experimental design with blinding, placebo control, repeated-measures within the same subjects, an a priori defined experimental plan (included as a supplementary document) and power calculation, using an established attentional analgesia paradigm that has been developed over a series of recent studies combined with sophisticated imaging protocols. This has allowed us to hold many of the experimental parameters constant (as indicated by the excellent reproducibility of the results across scan sessions and indeed corroboration with our previous studies) enabling the identification the effects of the drugs on attentional analgesia. We have transparently reported these findings for the reader, and they have both biological plausibility, reproducibility and are underpinned by statistical inferences. Therefore, we do not agree that it is correct to report that the lack of a 3-way interaction indicates a lack of influence of naltrexone on attentional analgesia.

Attentional effects in spinal cord responses. I am at a loss to follow the authors' claim in the abstract that "Noxious thermal forearm stimulation generated somatotopic-activation of dorsal horn (DH) whose activity correlated with pain report and mirrored attentional pain modulation.", as well as their lines 169-177, since the figure they added to their response letter shows exactly the opposite: there was no difference between HH and EH in the nociceptive cluster in the dorsal horn (I would actually find it very instructive for the reader if this figure were added to the supplement next to the same plot for the pain ratings; currently Figure 2 Supplement 2).

We see main effect of temperature in the spinal dorsal horn in the C6 segment which corresponds to the forearm dermatome where the thermal stimulus was applied (Figure 2 and also see the unmasked analysis – Figure 2- supplementary Figure 5) so we think the first part of this statement is amply justified *“Noxious thermal forearm stimulation generated somatotopic-activation of dorsal horn (DH)”*.

We provided additional data analysis after the previous round of revisions to show clearly that the parameter estimates extracted from this spinal_noci_ cluster correlate with pain scores across conditions with good coefficients (0.32-0.6) and slopes that were significantly nonzero so the second clause is supported *“whose activity correlated with pain report…”*.

This then leaves the statement *“mirroring attentional pain modulation”* which reflects the observation of the clear similarity between the plots of the behavioural pain scores across the four conditions and the parameter estimates extracted from the Spinal_noci_ cluster. We note the reviewers point about the lack of statistical significance which reflects the difficulty inherent in measuring small signal differences with fMRI in the spinal cord (as the reviewer is aware). The supplementary plot we have provided of this activity (Figure 2 supplementary figure 2B) shows the pattern of deltas in the means that we describe but also the large variance in the parameter estimates. For the purpose of the abstract we think *“mirrors”* is an appropriate concise term that does not imply statistical significance, we only ask the reader to compare the patterns shown in the two sets of graphs (Figure 2A and D) and they can draw their own conclusions. To enable this to be done we have included the requested plot in Figure 2- supplementary figure 2, have modified our statements in the results paragraph (p 6-7) and have added the following statement to the manuscript.

“Post hoc analysis of the differences in Spinal_noci_ BOLD in the hard|high – easy|high conditions, although showing the same pattern of differences in the means, did not show a group level difference between drug sessions. This absence of evidence for attentional modulation of absolute BOLD signal differences may reflect large interindividual differences, low signal to noise in spinal cord fMRI data, or an inability to discriminate between excitatory or inhibitory contributions to measured signal (Figure 2 -Supplementary figure 2B).”

Similarly, in lines 182-185 the authors make statements that are not based on any statistical inference.

At the reviewer’s request we have now conducted a post-hoc statistical analysis of the BOLD parameter estimates extracted from the spinal_int_ cluster. This shows that the spinal_int_ cluster is more active in the *high|hard* condition than in either the *easy|high* or *hard|low* conditions consistent with our proposal that this neural population may be an interneuron pool recruited in the *high|hard* condition. We further show that significant activation is seen in the spinal_int_ cluster in the *high|hard* condition in the presence of both placebo and reboxetine but not in naltrexone (Figure 2 supplementary figure 2C). These results have now been added to Figure 2E / legend and the results text and are noted in the discussion (p 12 para 2).

Assessment of group differences in connectivity. I appreciate the authors' response on this point, but it has unfortunately not assuaged my concerns at all. Since the authors use a 'pooled' analysis for the standard GLM analyses, why do they not use this 'pooled' approach for the PPI as well? This would mitigate the risk I have mentioned before, namely that the current approach might introduce a significant amount of bias into their results (e.g. partial 'double-dipping', see Kriegeskorte et al., 2009, Nature Neuroscience).

For the generalised psychophysiological interaction (gPPI) analysis we built upon our results from a completely independent sample (Oliva et al., 2021), to limit our seed/target regions to those previously found connected during attentional analgesia.

We hypothesised that, as a result of drug manipulation, changes in connectivity might be observed within this network. Given the inherent lack of statistical power in effective connectivity analyses (Friston et al., 1997; O'Reilly et al., 2012), we were concerned that initial pooling across the drug conditions might obscure the attentional analgesia network we sought to examine (if for example a drug altered connectivity). To this end, we focussed on the placebo (baseline) condition. The pattern of cortical/brainstem connectivity identified in this independent sample (N=39) is strikingly similar to that observed previously (N=57, (Oliva et al., 2021)), with the addition of the important new information about effective connectivity to the spinal cord (all assessed using *corrected statistics*, which is uncommon for reported PPI results). This should provide the reader with confidence that these results are (1) reliable and reproducible, and (2) do not introduced significant bias into the study.

Therefore, we reject the reviewer’s assertion that we should have initially pooled across conditions to determine the network involved in attentional analgesia. This could have altered the identified network, potentially eradicating the connection to the spinal cord – due to drug effects. We believe that we are testing our hypotheses appropriately, and our methods are described in enough detail to allow proper scrutiny by the readership of *eLife*. We do not believe that this approach has involved “partial double-dipping” of the sort described by Kriegeskorte et al., 2009 – indeed we have explicitly sought to avoid this kind of confound.

Spinal cord: choice of mask. I am still not able to follow the authors' rationale as to why they use rather unspecific vertebral level & hemi-cord masks instead of the available probabilistic masks, which represent the current gold-standard: i.e. why do they not limit their search to the segment of interest (C6) and the grey matter therein? I would ask the authors to indicate within the manuscript why they are using a mask that does not allow spatial assignment of BOLD responses with respect to segmental level and grey matter.

The masks the reviewer refers to as “rather unspecific”, were derived from the probabilistic atlases used in the Spinal Cord Toolbox, as such they reflect the least biased way to identify activity localised to a particular segment and level. Our demonstration that even when using an entire spinal cord mask we *still* find activity at the “correct” location, increases confidence in the result, not diminishes it (as the correction for multiple comparisons is more punitive). As an aside it should be noted that the probabilistic atlases are based on segmentations of high-resolution structural images, which are then transformed into the space of the spinal functional data (with their inherent distortions), so probably do not reflect the true extent of grey matter. Nonetheless we have clearly signposted our choice of probabilistic masks, and the rationale behind their use (see Page 23, Para 1).

Spinal cord: assessment of denoising success. It is unfortunate that the authors are not sharing the asked-for unmasked spinal cord data (but still use a cord mask), as such data would be most helpful for assessing the remaining level of physiological noise as well as the local nature of spinal cord BOLD responses (e.g. draining vein contributions). I am not able to follow the authors' argument that "due to masking steps in the registration pipeline, it was not possible to include tissues outside the spinal cord" – showing a few millimetres of data around the cord / CSF should be possible.

We have previously demonstrated the success of denoising techniques for identifying BOLD activity within the human spinal cord (Brooks et al., 2008; Brooks et al., 2012; Eippert et al., 2017; Kong et al., 2012). We took an identical approach in the current study and used what we had learnt from those methodological studies to help answer questions about spinal cord function and descending pain control, not the role of denoising in spinal fMRI. We have no reason at all to think that our data can be explained by some residual physiological noise artefact – the pattern of the activations at a spinal level is anatomically distinctive (reinforced by the unmasked analysis) and fits remarkably well with the known human and animal neurobiology.

References

Brooks, J.C., Beckmann, C.F., Miller, K.L., Wise, R.G., Porro, C.A., Tracey, I., and Jenkinson, M. (2008). Physiological noise modelling for spinal functional magnetic resonance imaging studies. Neuroimage 39, 680-692.

Brooks, J.C., Kong, Y., Lee, M.C., Warnaby, C.E., Wanigasekera, V., Jenkinson, M., and Tracey, I. (2012). Stimulus Site and Modality Dependence of Functional Activity within the Human Spinal Cord. J Neurosci 32, 6231-6239.

Eippert, F., Kong, Y., Winkler, A.M., Andersson, J.L., Finsterbusch, J., Buchel, C., Brooks, J.C.W., and Tracey, I. (2017). Investigating resting-state functional connectivity in the cervical spinal cord at 3T. Neuroimage 147, 589-601.

Friston, K.J., Buechel, C., Fink, G.R., Morris, J., Rolls, E., and Dolan, R.J. (1997). Psychophysiological and modulatory interactions in neuroimaging. Neuroimage 6, 218-229. Kong, Y., Jenkinson, M., Andersson, J., Tracey, I., and Brooks, J.C. (2012). Assessment of physiological noise modelling methods for functional imaging of the spinal cord. Neuroimage 60, 1538-1549.

Lakens, D. (2017). Equivalence Tests: A Practical Primer for t Tests, Correlations, and MetaAnalyses. Soc Psychol Personal Sci 8, 355-362.

O'Reilly, J.X., Woolrich, M.W., Behrens, T.E., Smith, S.M., and Johansen-Berg, H. (2012). Tools of the trade: psychophysiological interactions and functional connectivity. Soc Cogn Affect Neurosci 7, 604-609.

Oliva, V., Gregory, R., Davies, W.E., Harrison, L., Moran, R., Pickering, A.E., and Brooks, J.C.W. (2021). Parallel cortical-brainstem pathways to attentional analgesia. Neuroimage 226, 117548.